# Drosophila Topoisomerase 3β binds to mRNAs in vivo, contributes to their localization and stability, and counteracts premature aging

**Shohreh Teimuri, Beat Suter** *

Institute of Cell Biology, University of Bern, Berne, Switzerland

* Beat.Suter@unibe.ch

## Abstract

Topoisomerase 3β (Top3β) works not only on DNA but also on RNA. We isolated and identified the naturally cross-linked RNA targets of Drosophila Top3β from an early embryonic stage that contains almost exclusively maternal mRNAs. Favorite targets were long RNAs, particularly with long 3'UTRs, and RNAs that become localized in large cells. Top3β lacking only the hydroxyl group that makes the covalent bond to the RNA, did not allow normal expression and localization of Top3β mRNA targets or their protein products, demonstrating the importance of the enzymatic activity of Top3 β for optimized gene expression. *Top3β* is not essential for development to the adult stage but to maintain the morphology of the adult neuromuscular junction and to prevent premature loss of coordinated movement and aging. Alterations in human *Top3β* have been associated with several neurological diseases and cancers. The homologs of genes and (pre)mRNAs misexpressed in these conditions show the same characteristics identified in the Drosophila *Top3β* targets, suggesting that Drosophila could model human *Top3β*. An *in vivo* test of this model showed that the enzymatic activity of Top3β reduces the neurodegeneration caused by the cytotoxic human $(G4C2)_{49}$ RNA. *Top3β* supports normal gene expression, particularly of long and complex transcripts that must be transported and translationally controlled. These RNAs encode large cytoskeletal, cortical, and membrane proteins that are particularly important in large and long cells like motoneurons. Their reduced expression in the mutant seems to stress the cells, increasing the chances of developing neurodegenerative diseases.

## Introduction

Topoisomerase 3β (Top3β) is an RNA-binding protein (RBP) with dual enzymatic activity toward DNA and RNA [1,2]. Topoisomerases are better known for their activity toward DNA, where they resolve tension and supercoiling, a prerequisite for efficient transcription, transcription control, and replication [3,4] (recently also reviewed in [5]). Much less is known about the function of topoisomerases toward RNA. Along with other RBPs, Top3β interacts with the RNAi machinery [6], and it contributes to mRNA stability and translation,

**Data availability statement:** All relevant data for this study are publicly available from the NCBI Gene Expression Omnibus repository (https://www.ncbi.nlm.nih.gov/geo/query/acc.cgi?acc=GSE249429) or included as Supporting information.

**Funding:** This work was supported by the Swiss National Science Foundation project grants 31003A_173188 and 310030_205075 and by the Equipment grant 316030_150824 to BS. Support came also from the University of Bern to BS. The funders had no role in study design, data collection, and interpretation, or the decision to submit the work for publication.

**Competing interests:** The authors have declared that no competing interests exist.

particularly in the nervous system, which is significant for neurodevelopment and psychological and physical well-being [7–9]. Its functional importance is suggested by results that link impaired *Top3β* function to neurodevelopmental and cognitive disorders [7,10–12].

*Drosophila melanogaster* Top3β possesses the key features of a topoisomerase. The predicted catalytic tyrosine (Tyr, Y; [13,14] residue is present at position 332 (Y332). Top3β topoisomerases distinguish themselves from other topoisomerases by their conserved RNA-binding motif called RGG (Arginine Glycine Glycine) box, which binds to RNAs or proteins. Its paralogue Top3α, a DNA topoisomerase, possesses the catalytic tyrosine residue but lacks the RGG box [13]. *In vitro*, the RGG box is not required for the enzymatic activity but might play a role in binding to the RNA, thereby facilitating the enzymatic activity *in vivo* [7].

The neurological requirements for an RNA topoisomerase may, in part, reflect its activity on mRNAs that need to be transported over long distances to be translated at the right place and time. During such transport, long molecules, such as mRNAs, are likely to become entangled or suffer other structural problems that might interfere with their packaging, transport, unpackaging, and translation [1,15]. Therefore, a topoisomerase that acts on RNAs might be able to rescue such RNAs. As discussed by Su and colleagues [8], there is considerable evidence that Top3β activity on RNA has important functions in cells and that it might act at all levels of gene expression. Using human cell lines, these authors showed that Top3β indeed affects mRNA translation and RNA stability [8]. They also found evidence that these effects involve mainly TDRD3, which could be strengthened by the recently described mouse *Tdrd3*-null phenotype [9]. However, there is still surprisingly little knowledge about the steps in gene expression where Top3β acts on RNAs and particularly about the *in vivo* role of this activity for the cells and the organism. Especially in large cells, where mRNAs need to travel over long distances, an RNA topoisomerase could provide an important advantage and allow the cell to function properly. Drosophila has such large cells in the nervous system, the female germline, and the young, syncytial embryo. The latter is easily accessible and free of additional tissue. According to Flybase (https://flybase.org/reports/FBgn0026015), the expression of Drosophila Top3β is particularly abundant in 0–2 hours old embryos when the syncytial embryos are in their first 10 nuclear divisions cycles [16]. The mRNAs in these large cells are maternally provided during oogenesis and bulk zygotic transcription would only start about an hour later, around cellularizations (reviewed by [17,18]).

Proper mRNA localization to specific compartments combined with controlled local translation is essential for normal differentiation and cellular physiology and it has been studied extensively during these stages. This developmental stage, therefore, appears to provide a unique opportunity to study the activity and function of Drosophila Top3β toward cytoplasmic RNAs by identifying relevant *in vivo* RNA targets.

Here we show that in such 0–2 hours old embryos, longer RNAs were more frequently bound to Top3β⁺::GFP in a Y332-dependent way. This is consistent with the notion that larger RNAs are more likely to encounter structural problems that become targeted and repaired by the RNA topoisomerase in living cells. The same RNA features are also over-represented among the differentially expressed RNAs in the *Top3β* mutants. As opposed to capturing the targets in young preblastoderm embryos, the transcriptomics analysis of wild-type and mutant embryos derived from mothers with the respective genotype reveals the sum of direct and indirect effects of the cytoplasmic and the nuclear activities of Top3β. This information hints at the possible mechanisms that give rise to the mutant phenotypes. We found that *Top3β* is needed for the normal expression of a very large fraction of the mRNAs, but the lack of *Top3β* did not fully abolish their expression. Specific mutations of the Y332 active site tyrosine (Y332F) and the RGG box (ΔRGG) revealed the importance of their enzymatic and RNA/protein binding activity, respectively. The enzymatic activity of Top3β is needed to covalently

bind to mRNAs and for the normal accumulation of many long and complex mRNAs and their encoded proteins in general and also in specific subcellular locations. The lack of this enzymatic activity is not essential for the viability of the embryos or larvae but seems to stress the affected cells and causes neurodegeneration and premature aging of adults. Because the Drosophila enzyme affects the outcome of a human RNA that causes neurodegeneration, this Drosophila model produces also valuable novel insights into neurodegenerative diseases and cancers with which human and mice *Top3β* had been associated [1,4,7,11,12,19].

## Results

### *Top3β* modulates gene expression of a large fraction of the genome

To test whether Top3β affects gene expression at the RNA level, RNAs from 0–2 hours-old embryos were extracted from a wild-type control strain and the *Top3β* mutants *Top3β* $^{26}$, a null mutant [6,20] caused by a deletion of a large fraction of the coding sequence starting at codon Y141 and changing it into a stop codon, *Top3β* $^{Y332F}$, and *Top3β* $^{ΔRGG}$, point mutants generated for this work to test the function of Y332 and the RGG motives. All mutants were both maternally and zygotically mutant. Sequence analysis of the transcriptome carried out on biological triplicates established the effect of the different *Top3β* mutations on the RNA levels expressed in 0–2 hours old embryos (S1–S3 Tables, S1 Fig ). Using adjusted p-values (Adjp) <0.05 and log2 fold changes <−1 and >1, RNAs that showed level changes between the mutant and the wild-type strains were selected for a first analysis (Fig 1A, left and central panel). Much more stringent conditions (Adjp < 0.0002 and log2 fold changes <−1) were also used (Fig 1A, right panel). In both analyses, the strongest effect was observed with the null mutant *Top3β* $^{26}$, which also displayed a very strong reduction of the *Top3β* $^{26}$ mRNA levels (S1 Table, S1 Fig). As opposed to the null mutant, the mRNA with the point mutation was stable (S2 and S3 Tables). The gene ontology analysis of biological processes revealed similar results for *Top3β* $^{Y332F}$ and *Top3β* $^{26}$ (S2 Fig ) but no enrichment terms for *Top3β* $^{ΔRGG}$. The affected processes range mainly from morphogenesis, synapse signaling, and trans-membrane transport to cell adhesion and cuticle formation. Several of these processes are active repeatedly in the life cycle, and this includes neurons and their synapses.

Only a small fraction of the affected genes displayed upregulation in the mutants (Fig 1A, S3 Fig). The majority were downregulated. The downregulated genes still showed a high overlap between the point mutations and the null mutant, indicating that the point mutations act as partial loss-of-function mutations. 92% and 85%, respectively, of the effects seen with the Y332F mutant using the Adjp < 0.05 and Adjp < 0.0002, respectively, were also observed with the null mutant. This points to the importance of the active site Tyrosine (Y332) for the function of Top3β and indicates that the enzymatic activity is important for the proper expression levels of many genes. The strong overlap with the null mutant is also reassuring because the two mutants are from different genetic backgrounds. The *Top3β* $^{ΔRGG}$ mutation was induced in the same genetic background as the *Top3β* $^{Y332F}$ mutation. With both thresholds, only 11% of the genes expressed at lower levels in *Top3β* $^{Y332F}$ were also identified in the *Top3β* $^{ΔRGG}$ mutant. This shows that strain background differences do not, or only weakly, affect the data quality.

Large genes produce different mRNA isoforms more frequently, sometimes with different protein-coding capacities and even functions. With the standard expression profiling used so far, we might miss a strong effect on one isoform if other mRNA isoforms are not affected or compensate for the missing one. To find out whether *Top3β* is needed to produce normal levels of specific mRNA isoforms, we allocated the sequence hits to specific transcripts using the Salmon method [21] (S4–S7 Tables, Fig 1B). The expression levels of individual transcript isoforms were then compared between the null mutant and the wild type using as cut off

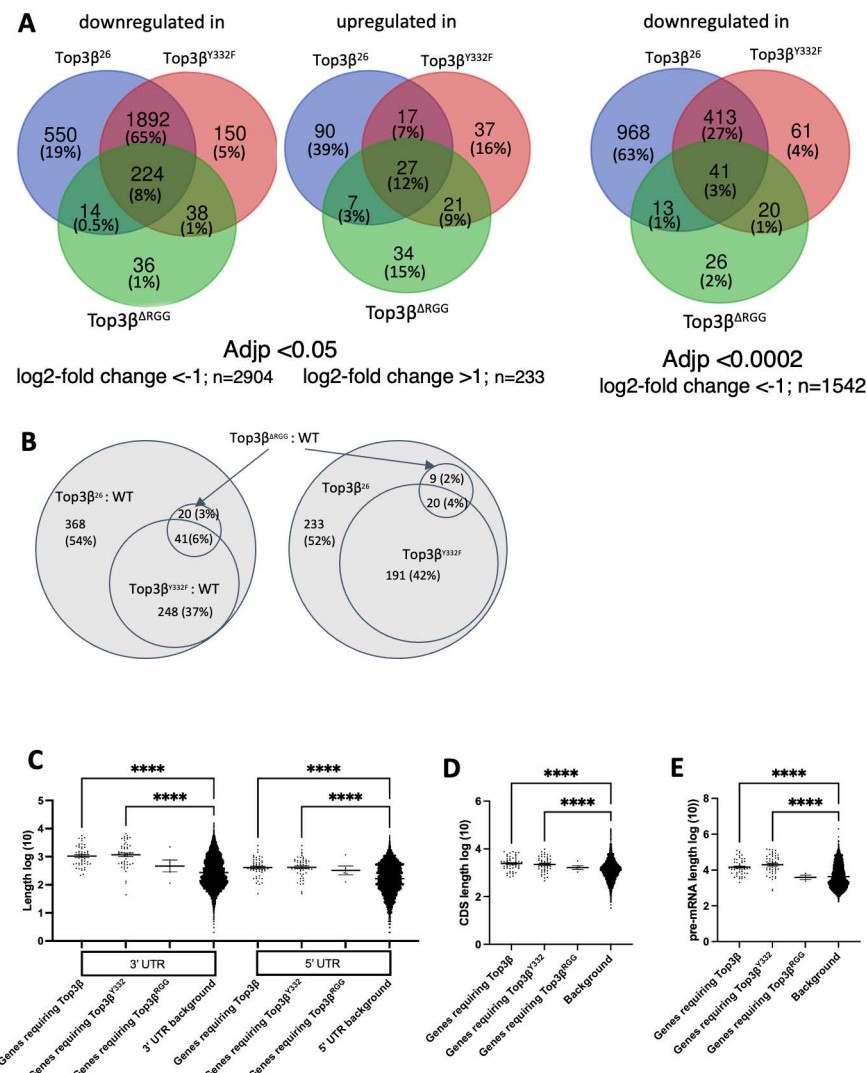

**Fig 1. Effect of the *Top3β* mutations on the transcriptome of 0–2 hrs. old embryos. A)** The left and central panels show the changes compared to the wild-type (WT) transcriptome using adjusted p-values (Adjp) < 0.05 and log2-fold change < −1 (downregulated) and >1 (upregulated). The Venn diagrams also show the allocation of the data to the 3 different mutants. Using the more stringent criteria Adjp < 0.0002 and log2-fold change < −1, the Venn diagram on the right shows the fractions of the genes with reduced transcript levels for the 3 alleles. n: is the number of different genes identified in the three mutants. **B)** Differentially expressed transcript isoforms in the *Top3β*[26] null mutant compared to the wild type (WT). Adjp < 0.05 and absolute log2-fold change ≥ 1. The Venn diagram shows which fraction is also differentially expressed in the point mutants when analyzed with the same stringency parameters (left panel). The fractions of the differentially expressed transcript isoforms expressed at lower levels in the mutants are shown in the right diagram. **C–E)** RNA features of the genes, which produce the transcript isoforms, that depend on *Top3β* to reach normal expression levels. **C)** Transcript isoforms from genes that produce long UTRs are over-represented in the group of RNAs that depend on *Top3β* and its Tyr[332] (Y332) residue to reach normal expression levels. **D)** Transcript isoforms from genes that encode longer CDS are expressed at higher levels in the wild type than in the *Top3β*[26] and *Top3β*[Y332F] mutants. **E)** Transcript isoforms from genes that produce longer pre-mRNAs are expressed at higher levels in the wild type than in the *Top3β*[26] and *Top3β*[Y332F]. Expression levels were compared to the *Top3β*+ (WT) control in 0–2 hrs. old embryos. p-value < 0.0001=****, p-value < 0.001=***, p-value < 0.01=**, p-value < 0.05=*. n = 50 for the null mutant *Top3β*[26] and *Top3β*[Y332F] and n = 5 for *Top3β*[ΔRGG].

Adjp < 0.05 and absolute log2 fold changes >1. This yielded 677 differentially expressed RNA transcript isoforms of which 67% (453) were downregulated (and 224 upregulated) in the null *Top3β²⁶* mutant, suggesting again that *Top3β* is mainly needed for the production or stability of RNAs. The transcript isoforms that are upregulated in the mutant might either be direct targets or generated by indirect, e.g., compensatory effects. This is possible because these mRNAs were produced during oogenesis in the nurse cells of the mother fly which carried the *Top3β²⁶* alleles. A closer inspection revealed that many genes with a transcript species that was upregulated in the mutant had a different isoform that was downregulated in the same mutant. This is consistent with a compensatory mechanism or a genomic background effect. Additionally, it could reflect the effect of lack of Top3β on splicing, polyadenylation site selection, or promoter usage, causing gene expression to shift to a different RNA isoform.

We next investigated the contribution of the catalytic Tyr, which is essential for the enzymatic activity, and the RGG motives, which mediate interactions with RNAs and proteins, toward the expression of specific transcript isoforms. To reduce strain-specific and random effects, we used the same cut-off values as before and focused on the 677 genes that are differentially expressed in the null mutant. As shown in Fig 1B, close to half (43%) of the transcript isoforms identified in the null mutant were also identified as differentially expressed in the *Top3β^{Y332F}* mutant (S7 Table). In the *Top3β^{Y332F}* mutant, 211 out of 289 (73%) differentially expressed RNAs were downregulated in the embryos, suggesting that the enzymatic function acts mainly to support RNA expression. Furthermore, this fraction is close to the 67% observed in the null mutant, indicating that the catalytic mutant has similar effects on the transcriptome of 0–2 hours old embryos as the null mutant. The impact of the mutated RGG domain on the embryonic transcriptome was less pronounced, with 29 out of 61 (47%) RNAs downregulated in the embryo, many might be indirect targets. Remarkably, 20 of these 29 RNA isoforms depended not only on the RGG motives of Top3β but also on the Y332 for their normal expression levels (Fig 1B).

0–2 hours old embryos do not measurably express zygotic genes yet but contain large quantities of mRNAs produced by the nurse cells in the ovary of their mother. Observed changes in their mRNA levels in *Top3β* mutants can, therefore, be caused by the effects of the mutations during transcription and RNA processing in the ovarian nurse cells of the mother, and during RNA localization and translation in the ovary of the mother or the young embryo. Furthermore, some of the observed effects are likely to be secondary effects, not directly caused by reduced *Top3β* activity.

We next selected from the list of preselected transcript isoforms (S7 Table) the 50 that showed the strongest reduction (according to their log2fc) in the null mutant and the Y332F mutant, and the top 5 for Δ RGG that were also in the top list of the null mutant. We then asked which RNA features were overrepresented in this group compared to the wild-type control. Selecting the top hits should reveal with higher confidence gene- and RNA features requiring *Top3β* activity to reach normal expression levels. On the other hand, many RNAs that appeared interesting according to the volcano blots were not included anymore (S1 Fig). In embryos, the RNAs that were less expressed in the *Top3β^{Y332F}* and *Top3β²⁶* mutants had longer untranslated regions (UTRs), protein-coding sequences (CDS), and pre-mRNAs (Fig 1C–1E). We conclude that long pre-mRNAs and mRNAs depend more strongly on *Top3β* to reach their normal expression levels.

The analysis of the significant changes in transcript isoform levels (S7 Table) revealed that at least 49 genes with reduced RNA levels of one isoform had at least one other isoform that was significantly higher expressed. Because these pairs might help to reveal the mode of action of *Top3β*, we studied 20 reciprocally expressed isoform pairs (S8 Table). In 11 cases, the upregulated transcript isoforms started more upstream, 3 further downstream, and 6 at the same

location. The latter group displayed alternative and additional splicing and 3'end formation. Top3β is a good candidate for modulating such splicing activities. Due to the different start locations observed in 14 cases, the pairs with the different transcription start sites differed in their 5'UTR and sometimes in the N-term of the ORF, too. Additionally, differences in splicing and splice site selection were also observed. Some pairs displayed different protein-coding capacities while others did not. The *Shaker cognate l-RA (ShaI-RA)* isoform was present at higher levels in the *Top3β* null and Y332F mutants, but contains a shorter last intron, causing earlier translation termination and lack of C-terminal parts of the voltage-gated K-channel. *Shal* is involved in neuronal physiology, locomotion, and lifespan. Another example of strong differences between the pairs of mRNA isoforms was found in *Tropomyosin-1 (TM1)*. Apart from 3'UTRs of different lengths, excision of a part of the 5'UTR, and different inclusions of small exons, the transcripts *TM1-RM* and *TM1-RA* start far apart and show very different alternative splicing patterns such that a large fraction of their ORF stems from different parts of the gene. Even though topoisomerases are involved in remodeling chromatin, the upregulation of transcript isoforms that initiate at different sites could also be the product of more indirect effects that compensate for the lack of another isoform.

### *In vivo* identification of cytoplasmic RNA targets of Top3β

Because Top3β is the only topoisomerase that also acts on RNA, we set out to specifically analyze its possible activities on cytoplasmic RNAs by capturing its RNA targets directly during the first 2 hours of embryogenesis. Because this time window ends before bulk zygotic transcription starts, we do not expect Top3β to bind to pre-mRNAs in this assay. For this experiment, the endogenous *Top3β* alleles were maternally and zygotically wild type (*Top3β$^+$*), and an additional copy of Top3β\*::eGFP was inserted on the third chromosome (\* stands for a wild-type or mutant allele). To identify directly the RNA targets to which Top3β binds and performs its topoisomerase activity, we sequenced the RNAs attached to Top3β$^+$::eGFP by immunoprecipitation (IP). This was done using animals that expressed an eGFP fused to wild-type Top3β and, as a negative control, Top3β$^{Y332F}$::eGFP in which Y332 was replaced by a phenylalanine (Y332F). This mutant form of Top3β lacks only the -OH group through which Top3β forms the transient covalent bond between its Y332 side chain and the RNA target during the enzymatic reaction [22]. We confirmed by qMS analysis that the wild-type and the mutant Top3β::eGFP fusions were expressed at similar levels (S9 Table). The naturally occurring covalent bond between the enzyme Top3β and its RNA targets makes artificial crosslinking superfluous while allowing the addition of ionic detergents at low concentrations during the isolation. Both measures contribute to reducing the background and pulling down primarily covalently bound RNAs without denaturing the antibodies and their interactions. We considered these conditions important to capture the interaction for two reasons. One, unlike many other RBPs, which are bound over an extended period to a large proportion of their target mRNA pool, we expected that only a small fraction of the potential targets would be bound to Top3β at the time of tissue lysis. Two, a large fraction of the maternally deposited RNAs is stored in an inactive form in the young embryo, preventing them from structural problems that would recruit the RNA topoisomerase. To preserve the covalent bond between Top3β's Y332 and the RNA targets after tissue lysis, we complexed Mg$^{2+}$, which is needed for the enzyme to resolve the bond at the end of the enzymatic reaction [3,14]. To determine if the target RNAs were specifically bound to the –OH group of Y332, we used the Y332F mutant as a negative control as described above. The RNAs that are enriched in the Top3β$^+$::eGFP (Top3β::eGFP) IP compared to the Y332F mutant are candidate targets of the catalytic activity of Top3β (S10 Table) because RNAs non-covalently interacting with Top3β will be washed off

the enzyme by the ionic detergent. Because long RNAs seem more likely to depend on this enzymatic activity, we further tested the quality of the data by comparing the RNA length of the presumed targets with the background RNA length. Indeed, longer RNAs were more likely to covalently bind to Top3β as they showed higher enrichment in IPs from embryonic extracts containing Top3β::eGFP compared to the Top3β[Y332F]::eGFP control (Fig 2A). It appears possible that a minor fraction of the suspected target RNAs represents false positives if their expression is upregulated by Top3β::eGFP but not by Top3β[Y332F]::eGFP expression. However, such an effect would not be expected to lead to the size correlation observed (Fig 2A). This argues that if such an effect exists, it would be only a minor one. The size correlation results might also reflect that longer RNAs have a higher chance of enduring physical stress during localization and translation and are, also for this reason, more likely to become substrates of the enzymatic activity of Top3β.

From the list of immunopurified RNAs, the ones with an adjusted p-value smaller than 0.05 (adjp-value < 0.05) and a log2-fold change of more than 1 (log2FC > 1) were selected and ranked based on their adjp-value. We used these parameters primarily because it is difficult to capture the

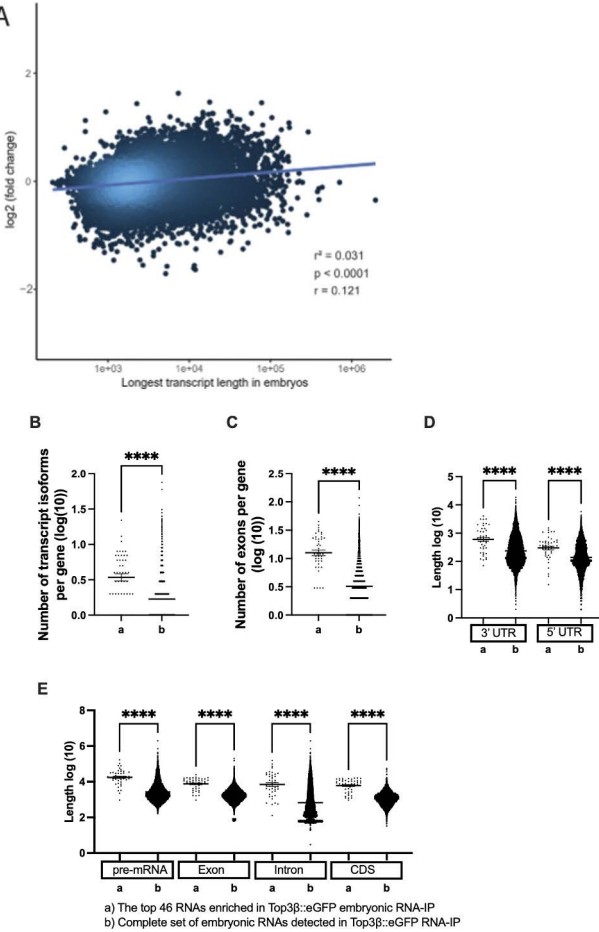

**Fig 2. Characteristics of embryonic RNA targets of Top3β::eGFP. A)** Longer RNAs are more likely targets of the enzymatic activity of Top3β because they are more enriched in the IP with the wild-type *Top3β* compared to the *Top3β*[Y332F]. r measures the strength of the correlation and p stands for p-value. **B–E)** Transcript features of the top 46 embryonic RNA targets of Top3β::eGFP. The complete sets of RNAs detected in the wild-type RNA-IP served as the standard. p-value < 0.0001=****, p-value < 0.001=***, p-value < 0.01=**, p-value < 0.05=*.

covalently bound phase. 46 embryonic RNAs and their genes fulfilled these criteria and were used for further analyses (S11 Table). The features that these RNAs share, can either indicate features that increase the chance of becoming a target of the enzymatic activity of Top3β or correlate with another feature that increases the chance of becoming a target. Drosophila embryonic Top3β RNA targets displayed on average a higher number of transcript isoforms and exons per gene, and longer 3'- and 5'UTRs compared to the complete set of different types of RNAs detected in the Top3β⁺::eGFP IPs (Fig 2B–2D , S10 and S11 Tables). The fact that targeted mRNAs have longer 3'UTRs is consistent with mRNAs that undergo localization and/or translational regulation having a higher chance of becoming a Top3β target because these processes are mainly controlled by sequences in the 3'UTR. A higher abundance of mRNAs with long coding sequences (CDS) might reflect that mRNAs with long open reading frames (ORFs) might become a Top3β target when their translation stalls (Fig 2E, S10 and S11 Tables). Whereas the longer introns and the higher number of exons might indicate that mRNA splicing increases the chance of becoming a Top3β target, the absence of transcription in the first two hours of embryogenesis seems to argue that these mRNAs are more likely to become targets because of other criteria that also apply to this group of genes. In summary, these results present evidence for the involvement of Top3β with long and complex mRNAs and mRNAs that are localized and/or under translational control. This is also consistent with a reported finding that long mRNAs are overrepresented among human mRNAs artificially crosslinked to Top3β in tissue culture cell lines [8].

## Top3β supports the expression and localization of its targets and their products

To evaluate the consequences of the interaction of Top3β with its RNA targets, we selected *shot* (encoding a spectraplakin), *Dhc64C* (encoding the large subunit of the cytoplasmic dynein motor), and *kst* (encoding the $\beta_{Heavy}$-spectrin) from the top list of RNAs covalently bound to Top3β::eGFP for further analysis. Aside from exhibiting the typical RNA features of the top targets, their homologs were also identified by the mammalian studies and they are expressed in neurons, too [1,23,24]. In normal young embryos, *shot* mRNAs accumulate more strongly in the polar granules and the pole cells at the posterior end (Fig 3). In embryos that were maternally and zygotically mutant for *Top3β*, *shot* mRNA levels were overall reduced. Additionally, the more intense, localized staining pattern in the posterior pole region was lost in the null- and the enzymatic mutant *Top3β^{Y332F}*, but not in *Top3β^{ΔRGG}*, suggesting that the normal expression and localization of *shot* mRNA depends on *Top3β* and particularly on its enzymatic activity. Analogous studies revealed that the anti-Shot antibody signal was similarly weaker in *Top3β^{26}* and *Top3β^{Y332F}* embryos compared to the wild type (Fig 3), suggesting that the reduced mRNA levels caused the reduced Shot protein expression. These findings are consistent with *shot* mRNA being a primary target of Top3β and they highlight the importance of the enzymatic activity of Top3β for the efficient expression and localization of the encoded protein. *Top3β^{ΔRGG}* mutant embryos still show a reduction of the Shot protein signal even though the RNA is still localized (Fig 3). This is consistent with the role of Top3β in the translation of *shot* RNA. That Top3β is required for efficient translation of mRNAs has already been demonstrated [8]. The effects on the proteins encoded by two more *Top3β* targets were similar. The cytoplasmic Dynein heavy chain (Dhc or Dhc64C) protein accumulated more strongly at the posterior end of the wild-type embryo, in front of the pole cells. This was neither detectable in the null mutant nor in *Top3β^{Y332F}* (Fig 3). The Kst protein accumulated at the apical cortex in normal blastoderm embryos, but in the *Top3β* null mutant and *Top3β^{Y332F}*, its signal was reduced throughout the embryo, including the apical cortex. Protein signals for Dhc showed an intermediate strength in the *Top3β^{ΔRGG}* mutant embryos (Fig 3), indicating

that the RGG box also affects their expression, albeit less strongly than the catalytic activity. We conclude that *Top3β* facilitates gene expression and particularly localized accumulation of proteins at sites distant from the nucleus.

Even though the lack of *Top3β* activity caused many changes in mRNA and protein levels and distribution, a strong phenotypic effect could not easily be identified in embryos. The reason for this might be that the lack of *Top3β* activity does not abolish target mRNA levels completely but only reduces them to different degrees. For most genes, the residual expression levels still permit basic activity or can be compensated to some degree, but this suboptimal physiology might cause cellular stress over time.

Top3β normally localizes to the same area in blastoderm embryos as the different proteins and mRNAs described here. In the Discussion section, we will detail how this seems to contribute to an interesting gene expression mechanism. Note also that the wild-type protein and

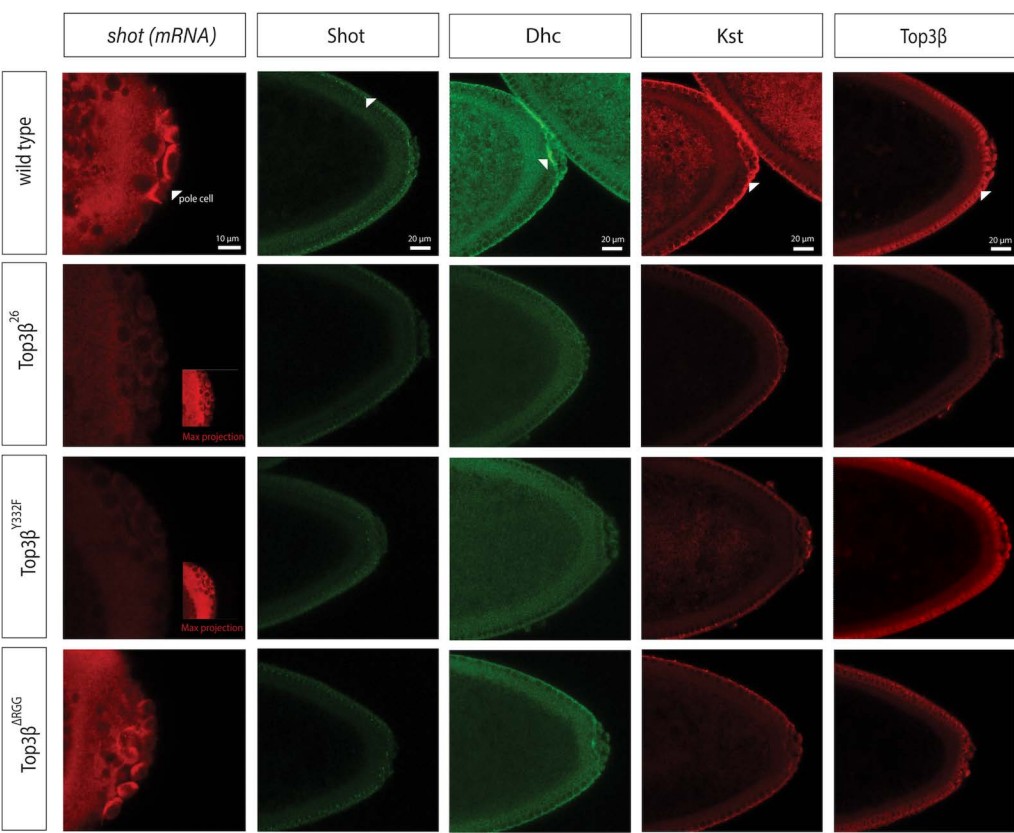

**Fig 3. Expression and localization of gene products of Top3β targets.** The posterior poles of the embryos are shown to the right. Comparing *shot* mRNA and Shot protein localization in syncytial (*shot* mRNA) and cellularizing (Shot protein) wild-type and *Top3β* mutant embryos shows lower signal levels and less localized signals in the posterior part of the embryos in the null mutant and *Top3β*$^{Y332F}$. Both were less reduced in *Top3β*$^{ΔRGG}$ mutant embryos. Note the *shot* mRNA localization in the pole cells in the wild type and *Top3β*$^{ΔRGG}$ mutant. Dhc64C localization at the posterior side of the somatic part of the embryo (arrowhead) was not detectable in the null mutant and *Top3β*$^{Y332F}$ and reduced in *Top3β*$^{ΔRGG}$ mutant embryos. Kst localization at the apical membrane (arrowhead) was reduced in null mutants and *Top3β*$^{Y332F}$ compared to the wild type. Top3β protein is localized similarly to its targets. It is, however, absent from the null mutant which reveals the background levels of the staining. Top3β is also present in the wild-type nuclei and in *Top3β*$^{Y332F}$ but absent from nuclei in the *Top3β*$^{ΔRGG}$ embryo. The same imaging settings were used for the same protein or mRNA staining in the different genotypes. Exceptions are the insets which were overexposed to reveal the embryo. The scale bar is 10μm for the *shot* mRNA pictures and 20μm for the other micrographs.

Top3β$^{Y332F}$ localize to the cytoplasm and the nuclei (Fig 3). In contrast, Top3β$^{ΔRGG}$ is seen above background levels only in the cytoplasm. This indicates that its RGG motives are needed to localize Top3β to the nucleus in blastoderm embryos and that the lack of the RGG box likely reduces the nuclear functions of Top3β$^{ΔRGG}$.

## Drosophila *Top3β* affects gene expression relevant to neurological diseases and cancers

Defects in *Top3β* cause reduced levels of many mRNAs, changes in isoform expression patterns (Fig 1, S1, S2, S4, S5 , S7 and S8 Tables), and reduced subcellular localization of the gene products (Fig 3) of many genes. We performed gene ontology analyses with the affected genes to identify physiological processes that might be affected later in the life cycle by reduced *Top3β* activity. The study of the Gene Ontology of Biological Processes suggested that *Top3β* might primarily affect neuronal processes from neuronal development to synaptic signaling and plasticity (S2 Fig). Human *Top3β* has been associated with several neurological diseases and cancers, too [1,4,7,11,12,19]. To test whether the Drosophila and human *Top3β* might similarly act to support the expression of higher levels, particularly of long RNAs, a list of diseases associated with *Top3β* was retrieved using gene-disease associations (DisGeNET (v7.0) [25]). The 12 diseases identified were either neurological diseases or cancers. The lists of differentially expressed genes associated with each disease were then obtained from the same database and used to identify their *Drosophila* orthologues from the DIOPT Version 9.0 (S12 Table). The list of these *Drosophila* genes was then compared with the list of transcript isoforms (and their genes) whose embryonic mRNA levels were reduced in the absence of *Top3β* (S7 Table). A sizable fraction of the 453 genes encoding *Top3β*-dependent isoforms, were also Drosophila orthologs of genes affected by these neurological diseases and cancers (Fig 4). Due to the small number of genes affected in the Juvenile Myoclonic Epilepsy (JME), this disease was not considered for further studies. For the remaining ones, there is a clear overlap between the two datasets, and the overlap ranges between 2 and >3 times the calculated random overlap, suggesting that the effects seen in the fly model with the inactivated *Top3β* also contribute to the human disease phenotype. The identification of neuronal mRNAs in 0–2 hrs old Drosophila embryos (i.e., before the nervous system is formed) is possible because many neuronal mRNAs are expressed already maternally and deposited into the egg for later use.

Notably, a single gene, *Rop* (*Ras opposite*), was present in the lists from all neurological diseases and the downregulated embryonic RNA isoforms (Fig 4C). *Rop,* is involved in vesicle release and control of synaptic activity [26,27], and *Rop* mRNA was identified as a Top3β target close to the cut-off with an adjp-value of 0.027 and a log2-fold change of 0.54 (S10 Table).

Comparing all cancer lists with the downregulated embryonic RNAs in the *Top3β* mutants revealed 50 genes shared between the six different lists (Fig 4D and 4E). Their KEGG pathway analysis indicated their involvement in the MAPK (mitogen-activated protein kinase) signaling pathway that regulates various cellular processes, including cell growth, proliferation, differentiation, and survival [28,29]. Thus, the MAPK signaling pathway seems particularly sensitive to *Top3β* inactivation.

We also compared the features of the Drosophila RNAs shared between the data sets in Fig 4A and 4B and compared them with the ones present only in the human disease-derived list or the fly list of embryonic RNAs that depend on *Top3β* (S4 Fig). Except for JME, the RNAs shared between the two datasets had on average longer 3' and 5' UTRs than the orthologs of the human disease RNAs that were not in the *Top3β*-dependent dataset. Similarly, in most

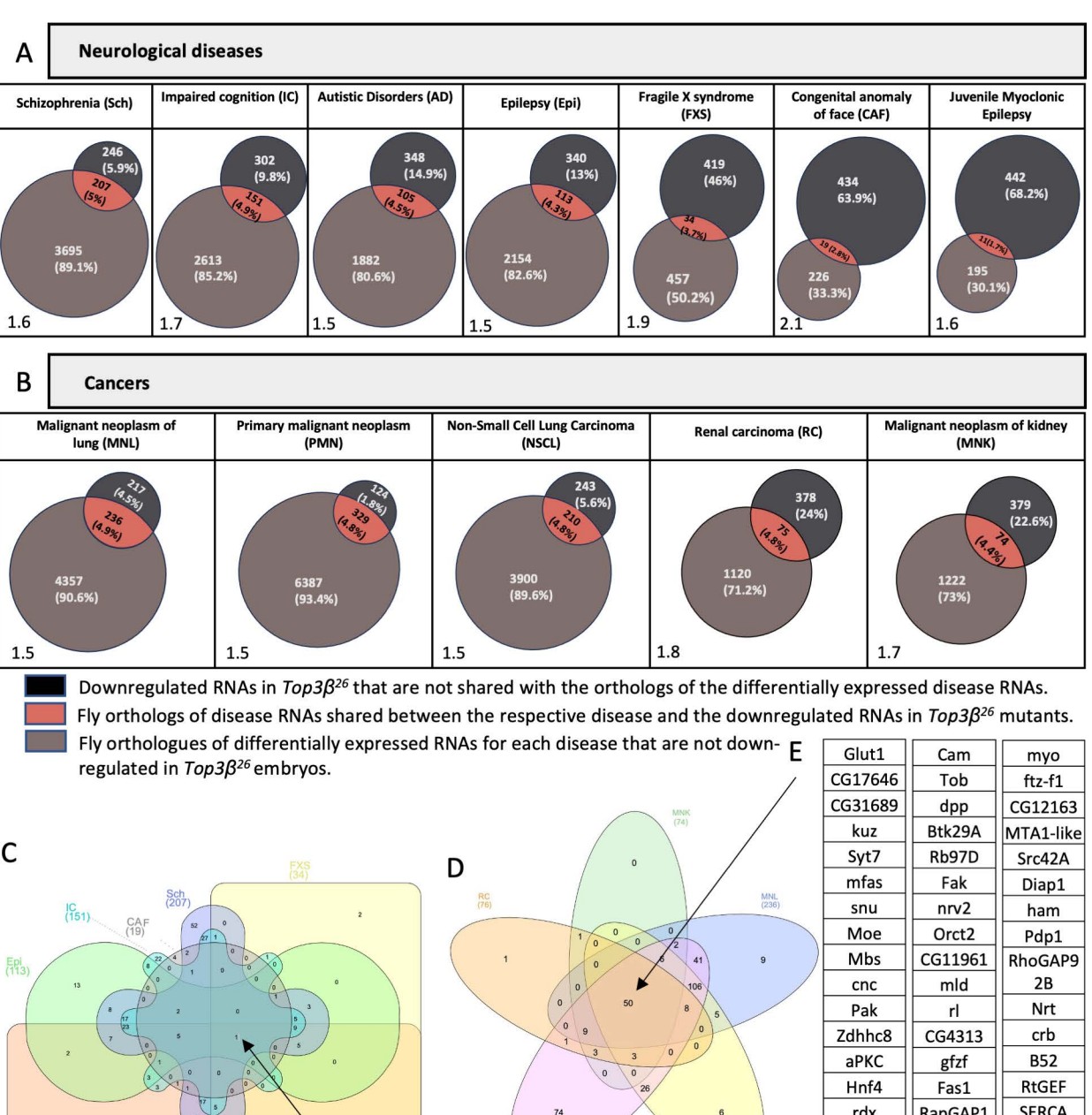

**Fig 4. Overlap between reduced embryonic RNAs in *Top3β²⁶* and the fly homologs of differentially expressed RNAs in several cancers and neurological diseases associated with human *Top3β*. A, B)** Venn diagrams of the genes shared between the list of reduced RNAs in *Top3β²⁶* embryos and the fly homologs of differentially expressed RNAs of the listed diseases. Numbers and frequencies are listed. The probability of random overlap is shown in % at the bottom left of each Venn diagram. **C)** *Drosophila* genes depending on *Top3β* for normal expression levels in embryos were placed into a Venn diagram such that their position shows in which neurological disease their homologs are differentially expressed (note that JME was not included). *Rop* appeared in all lists. **D)** *Drosophila* genes depending on *Top3β* for normal expression levels in embryos were placed into a Venn diagram such that their position shows in which cancer their homologs are differentially expressed. **E)** List of the 50 genes that depend on *Drosophila Top3β* for their normal expression levels in embryos and whose homologs are also affected in all listed cancers associated with human *Top3β* alterations.

cases, the pre-mRNAs in the overlapping group were significantly longer. This result is consistent with the human *Top3β* also affecting the expression of RNAs with the same features as its fly homolog, pointing to the potential value of identifying direct Top3β target mRNAs in the model system because their reduced expression might contribute to the disease phenotype.

## *Top3β* supports neuronal structure and function in aging adults

The similar molecular roles of human and Drosophila Top3β suggest similar requirements in the nervous system. The Drosophila negative geotaxis assay measures the activity of the nervous system and the muscles, and the coordination between them. In this assay, all four *Top3β* genotypes showed similar climbing abilities in young flies (Fig 5A). Neuromuscular conditions sometimes become only apparent at an older age because the young nervous system can compensate for some cellular deficits. However, such compensations can take their toll if they

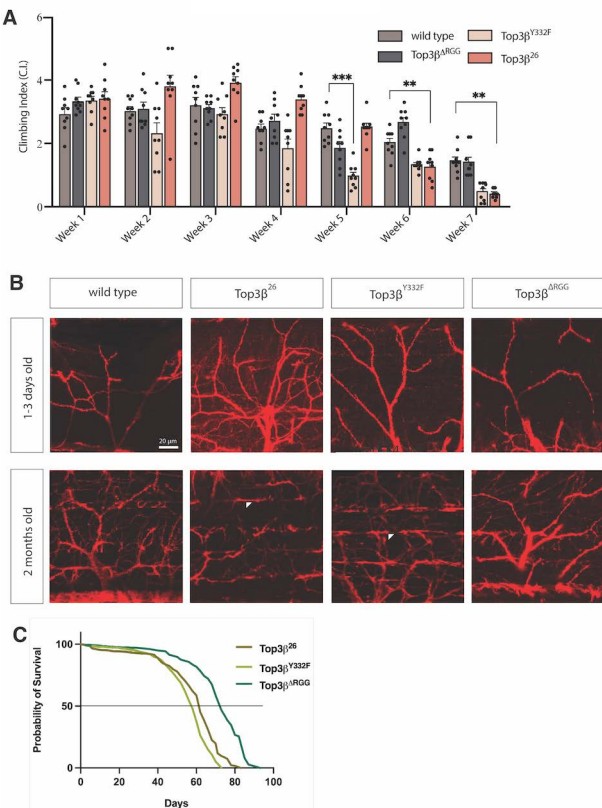

**Fig 5. Aging effects of *Top3β* mutants. A)** Climbing assays of wild type, *Top3β*^ΔRGG, *Top3β*^Y332F, and *Top3β*^26 flies. A two-way ANOVA analysis was performed. The graph bars show the mean climbing index (C.I.) for the different genotypes. Error bars represent the standard error of the mean (S.E.M). Each dot represents a calculated individual C.I. The adjusted p-values for each significant difference are indicated on the graph. (p-value < 0.0001=\*\*\*\*, p-value < 0.001=\*\*\*, p-value < 0.01=\*\*, p-value < 0.05=\*.) **B)** In adults, null mutant NMJs contain more branches than the wild type, indicative of a compensatory mechanism. Arrowheads point to the lost synaptic integrity of the dorsal longitudinal flight muscle (DLM). NMJ synapses were visualized by staining motor neurons with anti-HRP antibodies. The progressive denervation of the DLM is particularly evident in the null and the *Y332F* mutant in the adult thoracic NMJs. The scale bar represents 20µm. All panels show maximum intensity projections with a z-step size of 0.7µm. **C)** Life span shortening by *Top3β* mutations. *Top3β*^26 null and *Top3β*^Y332F mutants reduce long-term survival. *Top3β*^ΔRGG, which showed only a minor reduction compared to a wild-type strain in other experiments, served as a control because it was induced in the same background as *Top3β*^Y332F.

stress the cells. We, therefore, tested for premature aging and age-related degeneration of the climbing ability. Indeed, as seen in Fig 5A, from week 6 on, the $Top3\beta^{Y332F}$ and $Top3\beta^{26}$ flies became slower, and their climbing ability declined more than the wild-type one.

In aging flies with declining motor abilities and neurotransmission, fragmentation of synaptic motor terminals becomes apparent [30]. We studied the structure of the adult neuromuscular junctions (NMJs) in the four genotypes during aging (Fig 5B). Particularly the young null mutants contained more branched structures compared to the simpler branch structures in the wild type. This might constitute a mechanism to compensate for less efficient synapse function. Although to a lesser degree, young $Top3\beta^{Y332F}$ and $Top3\beta^{\Delta RGG}$ mutants also displayed presynaptic overgrowth. Concomitant with the premature loss of locomotive ability around week 6 (Fig 5A), denervation of the DLM (Dorsal Longitudinal Muscle) synapses became evident. Particularly in the null mutants, but also in $Top3\beta^{Y332F}$, loss of synaptic integrity was observed during aging in 1–2 months old adult flies (Fig 5B). The micrographs of these mutants are likely an underestimate of the average effect of the mutation because $Top3\beta^{26}$ and $Top3\beta^{Y332F}$ flies live less long than $Top3\beta^{\Delta RGG}$ (Fig 5C) and healthy flies, and only living flies were selected for the NMJ preparation. The decline of the NMJ structure and function was most likely more severe in the dead animals. In summary, it appears that the reduced expression of many Top3β targets can initially be compensated in neurons but over time leads to premature aging of the nervous system and reduced neuromuscular performance. Consistent with the identified Drosophila $Top3\beta$ phenotype, a shorter life span has also been reported for $Top3\beta^{-/-}$ mice [19].

## $Top3\beta$ counteracts the neuronal toxicity of the $(G4C2)_{49}$-RNA in flies

In addition to testing the effect of Top3β on the normal expression of cellular mRNAs, we tested whether Top3β's catalytic activity also affects the expression and the toxic effects of a repeat-expansion RNA, the human C9orf72 RNA. In humans, the presence of 20–28 repeats of this sequence was shown to be associated with the neurodegenerative diseases Amyotrophic Lateral Sclerosis (ALS) and Frontotemporal Dementia (FTD) [31–33]. Several laboratories established sensitive Drosophila systems where the effects of different genes and mutations on the phenotypes caused by the expression of this toxic RNA could be tested in neuronal cells that are not essential for viability. We used one of these systems with 49 repeat sequences $(GGGGCC)_{49}$ (also named $G4C2)_{49}$ [34] driven in the eye by the GMR-GAL4 driver [34] to study the effect of this RNA in the presence of the different $Top3\beta$ alleles. The very regular hexagonal patterns of the ommatidia allow the detection of neuronal cell death of only a few cells in the adult eye. Furthermore, because the eye is not needed for viability, effects can be analyzed even in the adult organism. $Top3\beta$ mutants enhanced the neurotoxic effect of the $(G4C2)_{49}$ transcripts in the eyes of young flies aged 1–2 days (Fig 6). Of the $Top3\beta$ mutants, $Top3\beta^{26}$ and $Top3\beta^{Y332F}$ exhibited the most severe eye lesions with irregular ommatidial structures, and this was particularly evident from the widespread cell death. Young $Top3\beta^{\Delta RGG}$ mutants displayed irregular ommatidia but usually less widespread cell death. After one month, the rough eye phenotype caused by the $Top3\beta^{26}$ and $Top3\beta^{Y332F}$ mutants affected the entire eye, and widespread degeneration of ommatidial cells was observed. These results show that the topoisomerase activity of Top3β counteracts the toxic activity of the $(G4C2)_{49}$ transcripts in the Drosophila model for this disease (Fig 6).

## Connecting the Drosophila Top3β data to mammalian Top3β, FMR1, and FMRP

The Drosophila proteins copurifying with Top3β::GFP support its suggested roles in translation, interacting with cytoplasmic RNAs and RNP granules (S9 Table, S5 Fig). The interaction

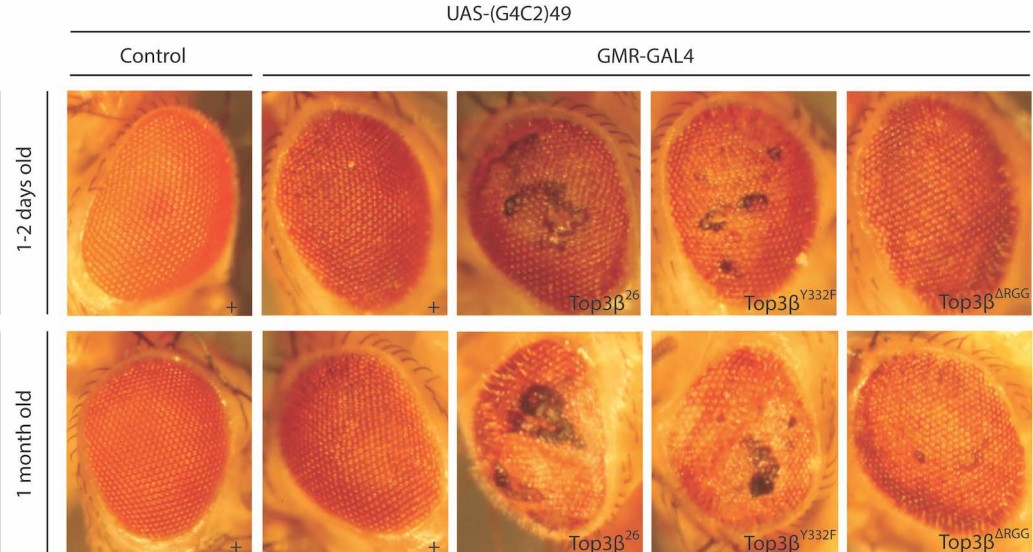

**Fig 6.** *Top3β* **counteracts the degenerative effect of (G4C2)**$_{49}$ **repeat RNAs in the eye.** Expression of the UAS-(G4C2)$_{49}$ transgene in wild-type (+) and *Top3β* mutant flies using the GMR-GAL4 driver. All *Top3β* mutants enhanced the degenerative phenotype of the (G4C2)$_{49}$ RNA. The *Top3β*$^{ΔRGG}$ allele caused mild disruptions, and *Top3β*$^{26}$ and *Top3β*$^{Y332F}$ mutants caused strong degeneration. The presented micrographs were from one set of experiments carried out in parallel; at least three replicates were performed. UAS-(G4C2)$_{49}$ and GMR-GAL4 lines were heterozygous. + indicates the control genotype *Top3β*$^+$; the mutant alleles are shown as superscripts.

network shows that one branch of its interactions goes through Tdrd3 and FMR1, the fly homolog of the fragile X mental retardation syndrome protein (FMRP) (S5B Fig). Top3β and FMR1/FMRP have both been linked to neurodevelopmental and neurodegenerative diseases (e.g., Fig 4) and functions in translation. For the latter role, it might for instance rescue entangled mRNAs in a way that they can be used for translation. Mammalian Top3β has been linked to FMRP, too, and it binds to ribosomes and regulates translation [1]. These researchers also found an overlap between HITS-CLIP data produced with FMRP and mouse brain extracts and Top3β HITS-CLIP data obtained with Flag-Top3β and HeLa extracts. To test the importance of the connection between Top3β and FMR1/FMRP, we compared our results with the published data of this interaction. Comparing the Drosophila Top3β::eGFP RNA-IP results with the shared mammalian result set, revealed that homologs of 12 of the 46 top maternal embryonic Drosophila targets were also present in the combined mammalian list. Considering that data not only from different species but also from different tissues were compared, this strong overlap reveals mRNA targets that support mainly cellular physiology in different cell types (Fig 7A). These results further strengthen the value of the technique applied to obtain the Drosophila Top3β target RNAs and the use of Drosophila as a model to study the role of Top3β in human disease. The common feature of the proteins encoded by the shared RNA targets is that they are predominantly large proteins. Many are localized to the plasma membrane or the cell cortex (e.g., Sev, Mgl, α-Spec, Kst, Pcx) or cytoskeleton-associated (e.g., Dhc64C, Mhc, Myo10A, α-Spec, Kst), have axonal functions (e.g., Shot, Dhc64C, Kst, Prosap), or scaffolding functions (Shot, Prosap). Consistent with similar results in vertebrates [8], these results show that there is a clear overlap between the Top3β and the FMRP activities but, additionally, the two proteins might perform activities independently of the other one.

We also compared the effects of Drosophila *Top3β* and *FMR1* on total RNA levels for each gene in 0–2 hrs old embryos. For this, we used the results from the *Top3β*$^{26}$ null mutant

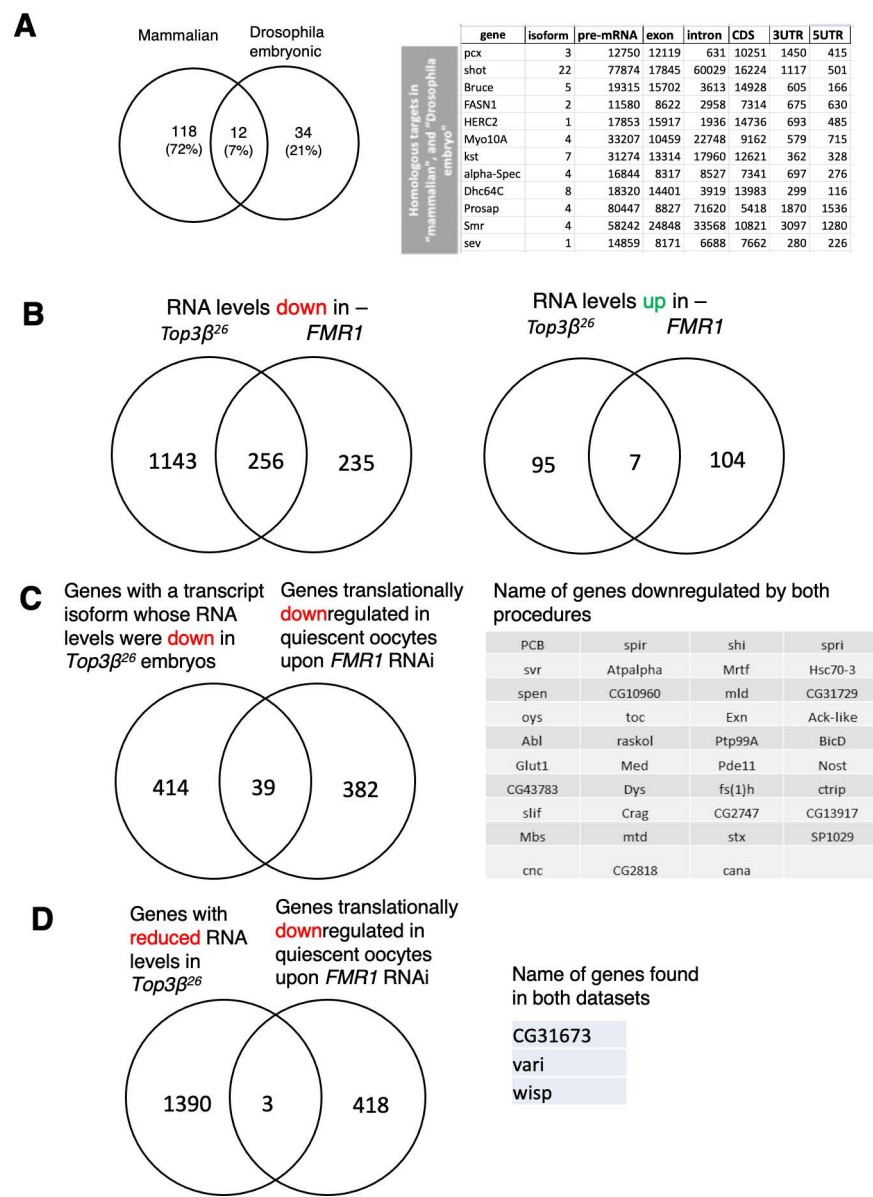

**Fig 7. Comparing Drosophila Top3β targets and effects to the corresponding FMR1/FMRP and mammalian Top3 β data. A)** A previous mammalian study produced a list of overlapping mouse FMRP targets in brain extracts and Top3β targets in HeLa cells [1]. The Venn diagram compares the list of Drosophila homologs of these mammalian targets to our list of direct targets of Top3β. The list of the Drosophila homologs that were also found in both mammalian interactor screens is shown with the characteristics of the Drosophila gene. The length measurements stem from the longest pre-mRNA isoforms (S11 Table). **B)** Comparing RNASeq data from 0–2 hrs old embryos from the *Top3β26* null mutant (Adjp < 0.0002; log2-fold change < −1 and >1, respectively, according to S1 Table) and the *FMR1* mutant [35]. The analysis considers reads per gene. Lists of the genes identified in the overlapping datasets are presented in the supplements (S13 Table). **C)** Genes encoding transcript isoforms that depend on *Top3β* for their normal accumulation in young embryos (S4 Table) and transcripts that depend on *FMR1* for their translation in quiescent oocytes [36]. The Venn diagram shows the overlap between the two datasets and the genes in the overlap are listed. **D)** Comparing RNASeq data from 0–2 hrs old *Top3β26* embryos (Adjp < 0.0002; log2-fold change < −1; S1 Table) and the *FMR1* RNAi effect on translational repression [36].

(Adjp < 0.0002; log2-fold change <−1 and >1, respectively; S1 Table) and the published results from an *FMR1* mutant [35]. These analyses consider reads per gene and do not discriminate between different RNA isoforms. Consistent with their physical interaction, more than half of the genes affected in the *FMR1* mutant were also found affected in *Top3β²⁶* (Fig 7B). In both cases, only a few genes showed higher RNA levels in the mutant and the overlap between the two elevated RNA datasets was very low. Lists of the genes identified in the datasets of both mutants are presented in the supplements (S13 Table).

The role of Drosophila *Fmr1* in translation was also studied in quiescent oocytes using RNAi treatment and analysis in ovaries [36]. Comparing the list of their genes significantly repressed in translation to the transcript isoforms less abundant in the extracts of *Top3β²⁶* mutants (S4 Table) revealed several shared mRNAs (Fig 7C). Even though different tissues had been used for the two experiments and different effects had been compared, there is still an overlap between the two data sets. Again, *FMR1* and *Top3β* seem to act on many of the same mRNAs, but they also seem involved in additional, independent functions. We also compared the effect of lack of Top3β on the RNASeq reads in 0–2 hrs old embryos per gene with the effect the RNAi treatment against *FMR1* has on translational repression [36] in the ovary. Using the data from S1 Table and the same conditions as in Fig 7B (Adjp < 0.0002; log2-fold change <−1) we only found three hits in the overlap with the same *FMR1* data (Fig 7D), which had shown a good overlap with the list of reduced transcript isoforms. Two reasons seem to explain the lower number in this overlap compared to the ones in **B)** and **C)**. One, as mentioned by the authors, *FMR1* RNAi knockdown did not generally reduce mRNA levels except for its own RNA [36]. In contrast, the *FMR1* mutant used in the study we analyzed for **B)** [35], reduced many RNA levels. Two, studying the effect on transcript isoforms (as in **C**) is more sensitive to detecting level changes because the same changes might be judged insignificant in the pool of different transcript isoforms of the same gene.

## Discussion

Top3β and its catalytic activity support the production of normal expression levels of a considerable number of RNAs in the developing embryo. Focusing on its activity towards cytoplasmic RNAs we presented the evidence for the important cytoplasmic functions of the Top3β topoisomerase in facilitating the translation of RNAs, translational control, and RNA localization. For instance, longer ORFs increase the risk of stalling translation because structural problems of mRNAs arise more frequently and prevent them from moving through the ribosome, causing their degradation if the structural problem cannot be solved. Top3β can rescue such mRNAs [8]. The entanglement of long mRNAs and pre-mRNAs poses a considerable problem during transport over extended distances. Some of these processes involve additional mRNA packaging and unpackaging before the mRNAs can be activated for translation at the right place and time [15]. The resulting structural problems might then also interfere with their transport. The coding sequence (CDS) and untranslated regions (UTRs) of mRNA molecules are important for regulating gene expression at the level of RNA localization and translation. Particularly 3'UTRs affect not only the translation but also the localization to the places where these mRNAs will be translated [15]. For example, in neurons, such mRNAs are localized in a translationally repressed form and become activated and translated in response to extrinsic stimuli, such as neurotrophins, synaptic activation, and regeneration after injury [28,37–39]. Consistent with the proposed function of Top3β in rescuing mRNAs with such functions and characteristics, the *in vivo* targets of Top3β display a strong bias for these characteristics (Fig 2, S11 Table) and the enzymatic activity of *Top3β* is particularly important for mRNAs with long 3'UTRs and CDSs (Fig 1B–1D, S7 Table). This proposed rescue function implies that Top3β is only needed for those mRNAs that run into structural problems that

would cause their degradation by the cells but not for "healthy" mRNAs. The results in S1 and S2 Tables and Fig 1 show that the absence of the RNA topoisomerase activity only reduces mRNA levels of affected RNAs but does not abolish them, which supports the proposed rescuing role for *Top3β*.

The top Top3β mRNA targets depended not only on *Top3β* for their normal expression levels, but also for their localization within the cell (Fig 3). RNA localization combined with translational control is a mechanism to provide local expression of gene products. It is most important in large cells, but also for fine-tuning gene expression in general. From our results, it seems likely that this lack of fine-tuning gene expression of numerous genes, particularly the ones with the characteristics of *Top3β* targets and *Top3β*-dependent transcripts, makes human *Top3β* mutations a risk factor for several neurological diseases and cancers.

Our main goal was to focus on Top3β's enzymatic activity toward cytoplasmic RNAs. However, the functional analysis of Top3β revealed also changes in mRNA isoform expression patterns that can be explained by the nuclear roles of Top3β in splicing, alternative splicing, and 3'end formation (S8 Table). This also fits the finding that Top3β's RNA targets often have a greater number of exons, longer introns, and alternative transcripts (Fig 2) and the protein interaction also revealed the interacting factors that could mediate these activities (S5 Fig, S9 Table). Future work should address whether Top3β interacts with these nuclear factors and the pre-mRNAs initially in the nucleus and gets exported similarly as other splice factors that remain associated with the spliced mRNA in the cytoplasm [40–44].

The results presented in Fig 1 also revealed that the null mutant affects many more genes that are not significantly affected by the lack of the enzymatic activity in the mutant Top3β$^{Y332F}$. Top3β, therefore, seems to play additional, more structural roles. This seems possible because many proteins still interact with Top3β in the Top3β$^{Y332F}$ and even in the Top3β$^{ΔRGG}$ mutant (S5 Fig, S9 Table). Some of these complexes likely mediate the non-enzymatic activities of Top3β that also affect RNA expression, turnover, and function. Su and colleagues [8] also reported that Top3β affects mRNA levels and translation by mechanisms that are dependent and independent of topoisomerase activity. Furthermore, Lee and colleagues [6] described the interaction of Drosophila Top3β with the RNAi machinery. Our complex analysis supported their results (S5 Fig, S9 Table). For the interaction with components of the RNAi-induced silencing complex (RISC), we found that Top3β interacts with Tdrd3-FMR1-Ago2 even without its Y332 and RGG motives, suggesting that even the enzymatically dead Top3β might affect the RNAi-induced silencing complex (RISC) in some ways.

Even though a high number of protein interactions with Top3β turned out to be independent of the RGG motives, we identified a non-enzymatic activity that depends on the presence of these RGG motives (Fig 3). Nuclear translocation of Top3β requires these RGG motives in the blastoderm embryo. The *Top3β*$^{ΔRGG}$ allele might thus be a useful tool to analyze the nuclear localization and functions of Top3β in the future. Interestingly, in HeLa cells, the RGG motives can be arginine methylated [45] and the methylation enhances the interaction with TDRD3, the topoisomerase activity, and stress granule localization upon arsenic treatment. Whether and how this methylation affects the nuclear localization of Top3β is not yet clear.

## Most important targets of Top3β?

*Rop* encodes a Sec-1-like protein involved in vesicle trafficking in neuronal synapses and non-neuronal vesicle trafficking. In neuronal tissues, it regulates neurotransmitter secretion in a dosage-dependent manner [26,27]. In *Rop* mutants, the release of neurotransmitters from the pre-synapse area into the synaptic cleft fails to occur. In 7 neurological diseases with which human *Top3β* had been associated and in Drosophila embryos, normal expression levels of the *Rop* mRNA depend on *Top3β* (Fig 4C, S7 Table). *Rop* is likely a rewarding gene to investigate

these neurological diseases. It was the only gene that was present in all 8 datasets and the *Rop* mRNA was also identified as a direct target of Top3β (S10 Table).

*shot* was identified in most experimental datasets analyzed. *shot* mRNA is a top Top3β target (S10 and S11 Tables) in Drosophila and HeLa cells and also as an FMRP target in mouse brains (Fig 7A; [1,8]). In young Drosophila embryos, normal levels of a specific *shot* mRNA isoform also depend on *Top3β* and its enzymatic activity (S5 Table) and *shot* mRNA and protein levels appeared reduced and less localized in embryos (Fig 3). *shot* encodes a spectraplakin family member, a large cytoskeletal linker molecule that binds actin and microtubules and participates in axon growth and neuromuscular junction maintenance and growth control [46–49]. Drosophila *shot* has additional roles in many other tissues and cell types, and it consists of 45 exons producing 22 transcript isoforms, as opposed to 2 isoforms per gene, which is average for *Drosophila*. The reduced *shot* mRNA and protein levels and localization might reflect the roles of Top3β in splicing *shot* pre-mRNAs, localizing *shot* mRNA, resolving RNA entanglements, and facilitating *shot* mRNA translation.

Localization of mRNAs combined with translational control is a mechanism that is particularly important in large cells (like oocytes and young embryos) and cells with large extensions (like neurons) [15]. Many of the identified mRNAs are needed in such cells and have important roles in neurons because they encode large cytoskeletal and cortical elements (e.g., *kst, α-spec, shot, mgl*) or regulators of NMJ growth (e.g., *Prosap*). Along with α-Spec, Kst ($β_H$-Spectrin) crosslinks F-actin and acts as a molecular scaffold when recruited to the apical membrane of epithelial cells [50,51] (Fig 3). Cortical localization seems to be one of the cellular processes that depend most heavily on RNA topoisomerase activity. For this reason, it is relevant that Top3β itself accumulates preferentially at the apical cortex of embryos, too. Higher Top3β activity in this location is expected to contribute to the apical translatability of mRNAs – for instance by rescuing entangled mRNAs -, thereby expressing the encoded protein apically at higher levels. Therefore, the subcellular localization of Top3β activity seems to serve as an additional tool to target gene expression efficiently to specific cellular locations. Such a mechanism can even produce protein localization if the mRNA is expressed uniformly.

## Top3β environment and relevance

The enrichment of neuronal-, stress-, and other ribonucleoprotein (RNP) forming granules, was a clear result of the analysis of the Top3β interactors (S5 Fig, S9 Table). In the case of the neuronal RNP granules, they concentrate specific sets of mRNAs and regulatory proteins in the same location, promoting their common long-distance transport to axons or dendrites [52,53]. They also have a dual function in regulating the translation of associated mRNAs [54]. As a component of neuronal RNPs, Me31B mediates the transport and controls the translation of neuronal RNAs, including the translational repression of synaptic transcripts [32]. We found Me31B highly enriched with Top3β and this interaction depended on the RGG box, suggesting that the RGG box can link Top3β to neuronal RNPs. It will be interesting to find out whether this localization is also controlled by Arg-methylation of these motives as reported for stress granule localization of Top3β [45].

The lack of RNA topoisomerase activity can interfere with the expression of many different genes involved in a variety of different pathways. The resulting suboptimal functioning of different cellular mechanisms is likely to interfere with cellular physiology, making it less efficient and producing cellular stress. Further research needs to test which of the individual pathways suggested by our work causes a physiological problem in the human conditions involving *Top3β*. Serious neurological health issues often arise when several pathways are not functioning properly anymore, making it difficult to pinpoint a single primary cause. Our work revealed that reduced activity of Top3β affects several pathways that rely on mRNAs that

are Top3β targets. The combination of defects in different physiological pathways is likely to impact the outcome of the various neurodevelopmental and cognitive disorders and cancers with altered *Top3β* function.

## Materials and methods

### Fly stocks and maintenance

The *white* (*w*) strain was used as a wild-type control. The mutant fly stocks *Top3β^Y332F^* and *Top3β^ΔRGG^* were established as described below. *Top3β^26^* was obtained from Lee and colleagues [6] and described by Wu et al [20]. The fly stocks were grown and maintained on a regular corn medium (ingredients: corn, yeast powder, sucrose syrup, agar, potassium sodium tartrate tetrahydrate, water, nipagin, and propionic acid) supplied with brewer's yeast for the experiments. All *Drosophila melanogaster* fly stocks were stored at 18°C in glass or plastic vials with day/night (12 h/12 h) light cycles. All experiments were performed at 25°C. $(G4C2)_{49}$ flies (Stock #84727; [34]) express 49 pure 5'GGGGCC3' repeats under the control of UAS. These repeats are neurotoxic and model the G4C2 repeats seen in human *C9orf72* that are associated with Amyotrophic Lateral Sclerosis and Frontotemporal Dementia.

### Generation of Top3β alleles

Genomic sequences were scanned for suitable sgRNA target sites with the JBrowse from FlyBase and appropriate sgRNA sequences were chosen based on (a) their distance from the tagging site, and (b) the absence of potential off-target sites on the same chromosome as the target site [55]. To mutate the endogenous *Top3β*, a sgRNA was designed, which hybridized in the vicinity of the Y332 codon of *Top3β* and induced the Cas9 nuclease to perform a double-strand break. Complementary oligodeoxynucleotides with suitable overhangs (S14 Table) were annealed and cloned into the pCFD5 vector, which is used to express the sgRNA [56]. The Cas9 fly stock was obtained from the Bloomington Stock Center (#79004). ssDNA was designed for the mutation changing the Tyr (Y) to a Phe (F) codon and injected into the progeny produced by Cas9- and sgRNA-expressing flies. The single-strand oligodeoxynucleotide with the flanking arms precisely matching the endogenous sequences changes the sequence at the mutation site by serving as a template (S14 Table) for the homologous recombination DNA repair at the double-strand break. The same strategy was used to delete the endogenous RGG region in *Top3β* (ΔRGG) using the appropriate gRNA and ssDNA templates. Stocks were established from individual mutants, and genomic sequencing confirmed the precise mutation events in the initial mutant fly and the final stock (S6 Fig). Stocks were deposited in the Bloomington Drosophila Stock Center (presently as #99728 and #99728). Both point mutants produce normal RNA levels (S1–S3 Tables) and also stable proteins (Fig 3) – although Top3β^ΔRGG^ was absent from the syncytial nuclei. As opposed to the null mutant, these two motive mutations did not seem to affect protein levels in embryos (Fig 3). qMS analyses of the tagged transgenic version of the different Top3β variants came to the same conclusion (see below).

### RNA sequencing

RNA was isolated using 1 ml of TRIzol® Reagent per 50–100 mg of tissue samples (0–2 hours embryos). The samples were homogenized at room temperature. RNA extraction was done according to the Trizol protocol. The University of Bern's Next Generation Sequencing Platform conducted quality control assessments, library generation, sequencing runs, and sequencing quality control steps. The purified total RNA was assessed for quantity and quality

using a Thermo Fisher Scientific Qubit 4.0 fluorometer with the Qubit RNA BR or HS Assay Kit (Thermo Fisher Scientific, Q10211 or Q32855, respectively) and an Agilent Advanced Analytical Fragment Analyzer System using a Fragment Analyzer RNA Kit (Agilent, DNF-471), respectively. rRNA depletion and construction of sequencing libraries were performed according to Lexogen's guidelines using a CORALL Total RNA-Seq Library Prep Kit with UDI 12nt Set A1 (Lexogen, 117.96). The libraries were paired-end sequenced using either shared Illumina NovaSeq 6000 SP, S1, or S2 Reagent Kits (200 cycles; Illumina, 200240719, 2002831, or 20028315) on an Illumina NovaSeq 6000 instrument. The quality of the sequencing runs was assessed using the Illumina Sequencing Analysis Viewer (Illumina version 2.4.7), and all base call files were demultiplexed and converted into FASTQ files using Illumina bcl2fastq conversion software v2.20.

The quality and quantity of reads generated from RNA-seq were evaluated. Most of the reads were from mature transcripts without introns, but we mapped them to the reference genome that includes introns using an alignment tool like Hisat2 that can handle large gaps. This required information on where each gene is located in the genome, available for example from Ensemble [57]. The number of reads mapping to each gene was counted using the FeatureCounts tool. This resulted in a table of read counts for each sample and gene. The bulk RNA seq data has been posted on NCBI-GEO (https://www.ncbi.nlm.nih.gov/geo/query/acc.cgi?acc=GSE249429).

To test for differential expression between two experimental groups containing three biological replicates each, we used the DESeq2 tool. The analysis included the following steps: 1. Normalization to correct for differences in the total number of reads between samples. 2. Variance estimation between replicates to account for the limited number of replicates in RNA-seq experiments (DESeq2 incorporates information from other genes with similar overall expression levels into the estimation). 3. Log-fold change (LFC) adjustment based on evidence strength which was estimated by the LFC. If it is weak (e.g., because the gene is expressed at low levels, the variance between replicates is high, or there are few replicates), the LFC shrinks toward zero. 4. Calculation of a test statistic and comparison to the normal distribution to obtain a p-value. 5. Multiple test corrections using the Benjamini-Hochberg procedure to consider only potentially detectable, differentially expressed genes. DESeq2 applies a false discovery rate correction based on the Benjamini-Hochberg procedure. However, the multiple-test correction considers only genes that could potentially be detected as differentially expressed. Only these genes will have an adjusted p-value. The mean read count across all samples is used to decide if a gene should be included or not. Mapping of the sequence reads to specific isoforms was performed with the Salmon method [21]. The Venn diagrams in Fig 1A were produced with a Ghent University online tool (https://bioinformatics.psb.ugent.be/webtools/Venn/).

## Extracting transcript features

The information about exon length, exon count, 3'UTR length, gene name, gene length, 5'UTR length, mRNA length, and CDS length was extracted from the.gtf file of dmel-all-filtered-r6.42.gff using a custom python script.

## Generation of the Top3β::eGFP cDNA clone for GAL4 > UAS expression

For the *Top3β::eGFP* construct, the *Top3β* cDNA clone LD10035 was amplified. Enhanced GFP (EGFP) was PCR amplified from a plasmid (pEGFP–C3). PCR amplified fragments were run on an agarose gel and purified using the Wizard SV kit (Promega). The sequences were cloned into the pUASz1.0 vector [58] with the *eGFP* ORF at the C-terminal end of the *Top3β* ORF using a ligation kit (NEB). The UASz vector allows one to induce the constructs

in any tissue, including the somatic tissue, the germline, and the CNS [58]. Plasmids were sequenced to confirm the integrity of the construct and purified with a PureYield kit (Promega) for injection. Transgenic stocks were established with the attP landing platform 86F from the Bloomington Stock Center (#24749) and also deposited in the Stock Center (#99726).

To generate an enzymatically dead enzyme, a Phe codon substituted the codon for the Tyr responsible for crosslinking the enzyme with the nucleic acid in the *UASz-Top3β::eGFP* vector to give rise to *UASz- Top3β$^{Y332f}$::eGFP*. Similarly, to figure out the role of the RGG box, we deleted in the *UASz-Top3β::eGFP* vector the sequence coding for this box, giving rise to *UASz-Top3β$^{ΔRGG}$::eGFP*. At the site of the desired mutation, oligonucleotide primers with the desired mutation were used to amplify the mutant double-stranded DNA plasmid (S14 Table). A protocol based on two different PCR amplifications was used for this. First, two elongation reactions were carried out, with one of the two primers each. In the second reaction, the two initial elongation reactions were mixed and an additional PCR reaction was carried out. The eGFP fusion constructs were transformed into *Drosophila* using the attP-86F landing platform. They are also available from the Bloomington Stock Center (#99725 and #99727). mat-tub-GAL4 (#7062) was used to drive the expression in the older female germline. The wild-type and the mutant eGFP fusions were expressed at similar levels as determined by qMS analysis (S9 Table). UASz-eGFP transgenic flies, obtained from the Bloomington stock center, were used as negative controls.

## Sample preparation for mass spectrometry

Drosophila Top3β::eGFP was purified using the GFP tag and a mouse anti-eGFP antibody as described previously [59]. Protein G magnetic beads were washed 3x with PBS and incubated for 2 hours with an anti-GFP antibody. After this, they were washed 3x with 1 ml 1x PBS and one more time with 1 ml homogenization buffer (25 mM Hepes PH 7.4; 150 mM NaCl; 0.5 mM EDTA PH 8.0, 1 mM DTT, protease inhibitors). To prepare the extracts, 1 ml homogenization buffer was added to 1 gr of the samples (dechorionated embryos 0–2 hours) in the glass homogenizer. The samples were homogenized and transferred into the tube. Homogenized samples were centrifuged at 16,000 g (RCF), 4°C for 40 min. The interphase was transferred into a fresh Eppendorf tube, centrifuged again at 16,000 g, 4°C for 40 min, and transferred into a fresh tube. Beads were then incubated with the extracts for 6 hours at 4°C on a wheel and the extracts were subsequently removed from the beads. The beads were washed once with 1 ml homogenization buffer and then several times with wash buffer (25 mM Hepes PH 7.4; 150 mM NaCl; 0.5 mM EDTA PH 8.0; 1 mM DTT; protease inhibitors). The wash buffer was removed, and the beads were sent for mass spectrometry.

## Mass spectrometry

Mass spectrometry analysis was done by the Proteomics and Mass Spectrometry Core Facility of the University of Bern, and the data are available in S9 Table. Proteins in the affinity pull-down were re-suspended in 8M Urea/ 50 mM Tris-HCl pH8, reduced for 30 min at 37°C with 0.1M DTT/ 100 mM Tris-HCl pH8, alkylated for 30 min at 37°C in the dark with IAA 0.5M/ 100 mM Tris-HCl pH8, dilute with 4 volumes of 20 mM Tris-HCl pH8/ 2 mM CaCl2 before overnight digestion at room temperature with 100 ng sequencing grade trypsin (Promega). Samples were centrifuged, and a magnet holder trapped the magnetic beads to extract the peptides in the supernatant. The digests were analyzed by liquid chromatography (LC)-MS/MS (PROXEON coupled to a QExactive mass spectrometer, ThermoFisher Scientific) with three injections of 5 µl digests. Peptides were trapped on a µ Precolumn C18

PepMap100 (5 μm, 100 Å, 300 μm × 5 mm, ThermoFisher Scientific, Reinach, Switzerland) and separated by backflush on a C18 column (5 μm, 100 Å, 75 μm × 15 cm, C18) by applying a 60 min gradient of 5% acetonitrile to 40% in water, 0.1% formic acid, at a flow rate of 350 ml/min. The Full Scan method was set with a resolution at 70,000 with an automatic gain control (AGC) target of 1E06 and a maximum ion injection time of 50 ms. The data-dependent method for precursor ion fragmentation was applied with the following settings: resolution 17,500, AGC of 1E05, maximum ion time of 110 milliseconds, mass window 2 m/z, collision energy 27, underfill ratio 1%, charge exclusion of unassigned and 1+ ions, and peptide match preferred, respectively.

The data were then processed with the software MaxQuant [60] version 1.6.14.0 against the UniProtKB [61] Drosophila 7228 database (release 2021_02) containing canonical and isoform entries, to which common contaminants were added. The following parameters were set: digestion by strict trypsin (maximum three missed cleavages), first search peptide tolerance of 15 ppm, MS/MS match tolerance of 20 ppm, PSM and protein FDR set to 0.01 and a minimum of 2 peptides requested per group. Carbamidomethylation on cysteine was selected as a fixed modification; the following variable modifications were allowed: methionine oxidation, deamidation of asparagines and glutamines, and protein N-terminal acetylation; a maximum of 3 modifications per peptide was allowed. Match between runs was turned on (match time windows 0.7 min) but only allowed within replicates of the same kind.

Peptides were normalized by variance stabilization [62], imputed and combined to form Top3 [63] intensities, and considered alongside MaxQuant's Label-Free Quantification (LFQ) values. Missing peptides, respectively LFQ intensities, were imputed by drawing values from a Gaussian distribution of width 0.3 centered at the sample distribution mean minus 2.5x the sample standard deviation, provided there were at most 1 non zero value in the group; otherwise, the Maximum Likelihood Estimation [64] was used. Differential expression was performed by applying the empirical Bayes test [65] between groups; significance testing was performed as described [66].

## RNA immunoprecipitation

Protein G magnetic beads were washed 2–3× with 1ml blocking buffer (20 mM Hepes, 150 mM KCl, 20% Glycerol, 0.5% Tween, BSA (Biolabs), Heparin 0.2 mg/ml, SDS 0.1%, 1/2 pill EDTA free protease inhibitors). Beads were blocked with a blocking buffer for 2–3 hours at room temperature. Anti-GFP antibody was added to the beads and rotated for 1–2 hours. Beads were washed 3x with non-hypotonic lysis buffer (20 mM Hepes, 1 mM EDTA, 150 mM KCL, 1 mM DTT, 0.5% Tween, 20% glycerol, SDS 0.1%, 1/2 pill EDTA free protease inhibitors).

Dechorionated embryos were 0–2 hours old. 1g of the embryos were homogenized in 2 ml hypotonic lysis buffer (20 mM Hepes, 10 mM KCL, 1 mM EDTA, 1 mM DTT, 0.5% Tween, SDS 0.1%, 1/2 pill EDTA free protease inhibitors, and RNase inhibitor (Biolabs, 40,000U/ml)). 600 μl of extract, 323 μl of Adjusting buffer (57% Glycerol 100%, 0.4M KCl, 1 mM EDTA, 1 mM DTT, 20 mM Hepes, SDS 0.1%), and 2 μl of RNase-free DNase I (20 μ/ml, Roche) were added to the washed antibody beads and were precipitated ON rotating at 4°C. Samples were washed 8x with 1 ml high salt wash buffer (20 mM Hepes, 1 mM EDTA, 200 mM KCl, 1 mM DTT, 0.5% Tween, 20% Glycerol, SDS 0.1%, 1/2 pill EDTA free protease inhibitors, and RNase inhibitor (Biolabs, 40,000U/ml)). On the last wash, 100 μl of the samples were taken before centrifugation. 100 μl of proteinase K buffer, 1.5 μl proteinase K (20 mg/ml Roche), and RNase inhibitors were added to each tube and were incubated for 30 minutes at 55°C. 1 ml Trizol reagent was added to each tube, and RNA was extracted according to the Trizol protocol (TRIzol™ Reagent, Invitrogen™, Catalog number: 15596026).

The Next Generation Sequencing Platform, University of Bern, conducted quality control assessments, library generation, sequencing runs, and sequencing quality control steps. The quantity and quality of the purified total RNA were assessed using a Thermo Fisher Scientific Qubit 4.0 fluorometer with the Qubit RNA BR or HS Assay Kit (Thermo Fisher Scientific, Q10211 or Q32855, respectively) and an Advanced Analytical Fragment Analyzer System using a Fragment Analyzer RNA Kit (Agilent, DNF-471), respectively. Sequencing libraries were made using a CORALL Total RNA-Seq Library Prep Kit, Version 1 with UDI 12nt Set A1 (Lexogen, 117.96) according to Lexogen's guidelines for this kit. For the library preparation, purified RNA was neither depleted for rRNA nor selected for poly(A+) RNA but used directly for reverse transcription with random primers. Pooled cDNA libraries were paired-end sequenced using either shared Illumina NovaSeq 6000 SP, S1, or S2 Reagent Kits (200 cycles; Illumina, 200240719, 2002831, or 20028315) on an Illumina NovaSeq 6000 instrument. The quality of the sequencing run was assessed using Illumina Sequencing Analysis Viewer (Illumina version 2.4.7) and all base call files were demultiplexed and converted into FASTQ files using Illumina bcl2fastq conversion software v2.20.

The quality of the RNA-seq reads was assessed using fastqc v0.11.9 [67] and RSeQC v4.0.0 [68]. Unique molecular identifiers (UMI) were extracted with umi-tools v1.1.2 [69], and adapters were trimmed with cutadapt v3.4.1 [70] according to manufacturer instructions. The reads were aligned to the Drosophila reference genome (BDGP6.32) using hisat2 v2.2.1 [71]. FeatureCounts v2.0.1 [72] was used to count the number of reads overlapping with each gene as specified in release 103 of the Ensembl genome annotation. The Bioconductor package DESeq2 v1.36.0 [73] was used to test for differential gene expression between the experimental groups. To generate the log2 fold change versus transcript length plots, the estimated log2 fold change per gene was plotted against the most extended transcript length per gene with R v4.2.1 [74]. We used the log2 fold change ≥1 and the adjusted p-value (adjp-value) <0.05 of enrichment over control for further analysis.

## Immunolocalization and FISH

For the immunostaining of adult Dorsal Longitudinal Muscles (DLMs), thoraces of 0–3 days old and 2 months old adult wild-type, $Top3\beta^{Y332F}$, $Top3\beta^{\Delta RGG}$, and $Top3\beta^{26}$ flies were separated from the rest of the body in 1x PBS. The thoraces were fixed in PFA (32% formaldehyde diluted to 4% with 1x PBS) for 30 min at room temperature and then washed 3x with PBS. All 1x PBS was removed from the tubes using a Pasteur pipette. The tubes were submerged into a liquid nitrogen flask for 10 seconds using cryogenic tweezers. Then, 300 µL of ice-cold 1x PBS was added to the samples. The thoraxes were cut into two hemithoraxes, transferred into 1x PBS [30], and blocked in 1x PBS with 0.1% normal goat serum and 0.2% Triton X-100 at pH 7.4 for at least one h at 4°C. Mouse anti-HRP or anti-Brp (nc82; 1:125 dilution) from Developmental Studies Hybridoma Bank were used to stain motor neurons. Primary antibodies were removed on the following day and the tissue was washed four times with PBST for 5 min each on a rotator. For the secondary antibody, goat anti-mouse A647 (1.5 mg/ml, diluted 1:200 dilution; Jackson ImmunoResearch Laboratories Inc.) was added to the samples, which were then kept at room temperature for 2 hours in a dark box on the rotator. After the 2 h incubation, the secondary antibodies were washed out 4 times for 5 min with PBST. Then, the samples were mounted on a slide [30].

Embryonic RNA in situ hybridizations (FISH) and antibody staining techniques, respectively, were carried out as described previously in references [75] and [76], respectively. Anti-βH-Spectrin/Kst antibody (no. 243 used at 1:500–1:600) was obtained from [77], anti-Shot (mAbRod1) and anti-Dhc (2C11-2) antibodies were obtained from the Developmental Studies Hybridoma Bank (https://dshb.biology.uiowa.edu).

## Negative geotaxis (climbing) assays

Wild-type flies and the three *Top3β* mutant strains were reared on fresh food and allowed to hatch at 25° C. For each genotype, all 0-3-day-old adult male flies were collected. Ten young adult male flies, each, were anesthetized and placed in 4 separate graduated flat-bottom glass cylinders. After 1 hour of recovery from $CO_2$ exposure, flies were gently tapped 4x to the bottom of the cylinder to startle them repeatedly with constant force to displace the flies to the bottom surface. The flies were then allowed to climb the wall until they reached the top of the column. The climbing procedure was video recorded using an iPhone 11 Pro. These assays were repeated every week over 7 weeks on the same weekday. Three fly batches, each, provided the material for the triplicates which were performed within 5-minute intervals. After the three trials, each batch of 10 flies was transferred into glass vials and kept at 25° C. Before the experiment, every single batch was inspected to ensure that all 40 flies could be video-tracked when they started to climb after the tapping. To calculate each climbing index (CI), flies that remained at the bottom were counted as in zone 0. The formula used to calculate the CI is $CI = (0 \times n0 + 1 \times n1 + 2 \times n2 + 3 \times n3 + 4 \times n4 + 5 \times n5)/nTotal$, where nTotal is the total number of flies and nx is the number of flies that reached zone x.

## Statistics

All statistical analyses were performed using GraphPad Prism 8. For the negative geotaxis assays, the means, standard error of the means, and p-values were obtained by two-way ANOVA with Tukey's multiple comparison test.

## Supporting information

**S1 Fig. Transcriptomics of 0–2 hrs embryos.** Volcano plots showing the effect of three *Top3β* mutations on the transcript levels in 0–2 hrs old Drosophila embryos. Adjusted p values are plotted against differences in transcript abundance. The logarithmic scales used for both axes are indicated.
(TIF)

**S2 Fig. Gene ontology biological processes.** The enzymatic dead mutant *Top3β*[Y332F] and the null mutant *Top3β*[26] show very similar effects on biological processes, indicating that the Tyr in the active site is crucial for the role of Topβ in the expression of normal mRNA levels for these Biological Processes. The *Top3β*[ARGG] mutant did not reveal enrichment terms.
(TIF)

**S3 Fig. RNA levels elevated in the *Top3β* mutants.** Genes showing higher transcript levels in 0–2 hrs old embryos mutant for *Top3β* (compared to their wild-type expression). Adip < 0.0002; log2 fold changes ≥ 1. The Venn diagram shows the pairwise overlap between the different mutants.
(TIF)

**S4 Fig. Transcript features for the differentially expressed RNAs from the embryo and the fly homologs of genes associated with the indicated diseases. A, B)** mRNAs shared between the two lists show generally longer UTRs compared to the fly orthologues of differentially expressed disease RNAs that are not downregulated in the *Top3β*[26] mutant embryos. **C)** CDS length revealed no significant difference between the three gene lists for each disease (except for AD). **D)** Shared mRNAs show generally longer pre-mRNAs. p-value < 0.0001=****, p-value < 0.001=***, p-value < 0.01=**, p-value < 0.05=*. Sch: Schizophrenia, IC: Impaired cognition, AD: Autistic Disorders, Epi: Epilepsy, FXS: Fragile X syndrome, CAF: Congenital

anomaly of face, JME: Juvenile Myoclonic Epilepsy, MNL: Malignant neoplasm of the lung, PMN: Primary malignant neoplasm, NSCL: Non-Small Cell lung carcinoma, RC: Renal carcinoma, MNK: Malignant neoplasm of the kidney.
(TIF)

**S5 Fig. Pathways and protein interactions of Drosophila Top3β. A)** Gene ontology enrichment for Top3β-associated proteins for the top 100 interactors identified in embryonic extracts. Immunoprecipitations (IP) on extracts from 0–2 hours old embryos were performed using *Top3β::eGFP* and the *eGFP* control line. Polypeptide components of the IP complexes were analyzed using mass spectrometry (MS), and their abundance was compared to the eGFP control. This resulted in a list of 426 potential complex components with an adjusted p-value <0.01 and log2FC >1 (S9 Table). 89 ribosomal proteins were in this set. A gene ontology enrichment analysis was then performed for the top 100 (according to log2FC) non-ribosomal proteins. This revealed proteins involved in the activation of translation, P-bodies, stress granules, and neural ribonucleoprotein granules, all membrane-less structures involved in storing specific mRNAs during periods of stress or concentrating mRNAs and regulatory proteins [78]. Additionally, the interactors were also enriched in several functions related to mitochondria. **B)** Proteins are enriched in the embryonic protein-IP with their physical interactions, according to Cytoscape-String. Proteins that ended up further away from Top3β than the ones shown here, were removed from the interaction map. Tdrd3, an established interactor of Top3β [79], was among the top enriched proteins, suggesting that our immunoprecipitation was specific. To assess the role of the Y332 residue and the RGG box in Top3β interactions, IP results were compared with the ones from the *Top3β* mutants *Top3β^Y332F^::eGFP* and *Top3β^ΔRGG^::eGFP*. Among the 337 non-ribosomal binding candidates, 102 needed the Y residue and the RGG box to bind to Top3β::eGFP (S9 Table). An additional 16 proteins needed the Tyr (but not the RGG) and 23 the RGG box (but not the Y332) for their binding to Top3β. A large fraction of the identified proteins is involved in RNA transport, translation, splicing, mRNA surveillance, and degradation. The physical interaction map of the proteins enriched in the Top3β::GFP IP was created using Cytoscape-String and the results from the mutant analyses were entered into this map. The resulting map resolved into two distinct submaps. The left branch connects Top3β functions to its nuclear functions and includes splicing factors. The right branch is enriched in cytoplasmic proteins and includes Tdrd3, FMR1, the piRNA pathway, and translation. The first associations with Tdrd3 and FMRP have already been described from work with vertebrate cell cultures [8]. The interaction of Me31B with Top3β turned out to be dependent on the RGG box. As an RNA helicase, Me31B is a core component of a variety of ribonucleoprotein complexes (RNPs) that participate in translational control and mRNA decapping during embryogenesis, oogenesis, neurogenesis, and neurotransmission [52,53,80–85]. Me31B RNPs also contain Tral, eIF4E1, Cup, and pAbp. They are also involved in RNA localization and translational control of maternal mRNAs during oogenesis and embryogenesis [84]. These other complex proteins were also enriched in the embryonic Top3β::eGFP IP.
(TIF)

**S6 Fig. *Top3β* Gene structure showing the primer locations, confirmation of targeted mutations, and effect of these mutations.**
(TIF)

**S1 Table. *Top3β^26^* transcriptome analysis.** This table lists the results of the transcriptome analysis of 0–2 hrs old *Top3β^26^* embryos and compares them to the wild-type transcriptome.
(XLSX)

**S2 Table. *Top3β^{Y332F}* transcriptome analysis.** This table lists the results of the transcriptome analysis of 0–2 hrs old *Top3β^{Y332F}* embryos and compares them to the wild-type transcriptome.
(XLSX)

**S3 Table. *Top3β^{ΔRGG}* transcriptome analysis.** This table lists the results of the transcriptome analysis of 0–2 hrs old *Top3β^{ΔRGG}* embryos and compares them to the wild-type transcriptome.
(XLSX)

**S4 Table. *Top3β^{26}* transcript isoform analysis.** This table lists the transcript isoforms that change their expression levels in *Top3β^{26}* mutant embryos compared to the wild type.
(XLSX)

**S5 Table. *Top3β^{Y332F}* transcript isoform analysis.** This table lists the transcript isoforms that change their expression levels in *Top3β^{Y332F}* mutant embryos compared to the wild type.
(XLSX)

**S6 Table. *Top3β^{ΔRGG}* transcript isoform analysis.** This table lists the transcript isoforms that change their expression levels in *Top3β^{ΔRGG}* mutant embryos compared to the wild type.
(XLSX)

**S7 Table. 0–2 hrs embryonic transcript isoforms affected by mutations in Top3β.** This file lists the filtered Top3β transcriptomics results for specific mRNA isoforms (with adjusted p-value < 0.05 and absolute log2 fold change >1). Columns C, D: Top3β^{26} vs wild-type embryos. Columns E, F: Top3β^{ΔRGG} vs wild-type embryos. Columns G, H: Top3β^{Y332F} vs wild-type embryos. Columns I: gene name (symbol), Columns J-M: length of indicated RNA features. The embryo background list (2^{nd} sheet) shows the length of the RNA features for all transcripts identified.
(XLSX)

**S8 Table. Reciprocal effects on the abundance of RNA isoform pairs in the *Top3β* mutants.** Selected pairs of transcript isoforms from the same gene show reciprocal expression levels in *Top3β^{Y332F}* and *Top3β^{26}*, but (with one exception) not in not *Top3β^{ΔRGG}*.
(XLSX)

**S9 Table. List of proteins co-immunoprecipitated (co-IPed) with Top3β::GFP.** Quantitative MS determined the enrichment in comparison to GFP-only IPs. co-IPs with mutant versions of Top3β*::GFP are also shown. They show the dependence of the interaction on the RGG domain and the Y332 residue.
(XLSX)

**S10 Table. List of RNAs (and gene names) that purified with Top3β::GFP from embryonic extracts.** Enrichment of RNAs in the Top3β::GFP IP versus the Top3β^{Y332F}::GFP mutant protein IP was used to select the best target RNAs. RNA features are indicated for these RNAs, too.
(XLSX)

**S11 Table. List of the top RNA targets of Top3β::GFP with their RNA features indicated.** Length measurements shown are derived from the longest pre-mRNA isoforms.
(XLSX)

**S12 Table. List of Drosophila genes whose human homologs are affected by the neurological diseases and cancers studied in** Fig 4**.**
(XLSX)

**S13 Table. List of genes affected in the RNAseq results from 0–2 hrs old Drosophila embryos mutant for _Top3β²⁶_ and _FMR1_.** All combinations of elevated and lower reads were considered. Genes with the following expression level changes were selected: adjusted p-value < 0.0002 and absolute log2-fold changes >1.
(XLSX)

**S14 Table. List of oligos, primers, and fly lines.**
(XLSX)

## Acknowledgments

Our thanks go to P. Nicholson and the members of the University of Bern NGS platform for sequencing the many RNA samples and to the IBU group of R. Bruggmann, particularly, S. Oberhänsli, G. van Geest, and R. Dörig for their important support during the analyses of the sequencing data and for providing the requested S6 Fig. We thank the University of Bern (Switzerland) Mass Spectrometry Center (PMSCF) members S. Braga Lagache and N. Buchs for performing the MS experiments and M. Heller and A-C. Uldry for the data analysis. We wish to acknowledge all our group members who gave their time and expertise to provide critical input and technical support. Special thanks go to P. Vazquez, R. Dörig, and D. Beuchle. We highly appreciate the continuous support from friends and colleagues in the Drosophila community for sharing their results, fly stocks, and antibodies. For this work, too, FlyBase provided invaluable support, the Developmental Studies Hybridoma Bank provided antibodies, and the Bloomington Drosophila Stock Center (NIH P40OD018537) fly stocks.

## Author contributions

**Conceptualization:** Beat Suter.

**Data curation:** Shohreh Teimuri.

**Formal analysis:** Shohreh Teimuri, Beat Suter.

**Funding acquisition:** Beat Suter.

**Investigation:** Shohreh Teimuri.

**Methodology:** Beat Suter.

**Project administration:** Beat Suter.

**Resources:** Beat Suter.

**Supervision:** Beat Suter.

**Validation:** Shohreh Teimuri.

**Visualization:** Shohreh Teimuri.

**Writing – original draft:** Shohreh Teimuri, Beat Suter.

**Writing – review & editing:** Beat Suter.

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
