## [Decision Letter · Decision Letter 0]

2 May 2024

PONE-D-24-09977RNA Targets and Physiological Role of Topoisomerase 3βPLOS ONE

Dear Dr. Suter,

Thank you for submitting your manuscript to PLOS ONE. After careful consideration, we feel that it has merit but does not fully meet PLOS ONE’s publication criteria as it currently stands. Therefore, we invite you to submit a revised version of the manuscript that addresses the points raised during the review process.

You will see that the two reviewers believe that your result provides new knowledge in the field of Top3b and how, in particular, it could be at the origin of neuropathologies.

But you will also see that they consider that the manuscript needs to be improved, in particular by carrying out additional experiments.

I broadly agree with their comments. I won't go into the details of the additional experiments requested (I know there are many), but I invite you to carry them out and, above all, to respond to all the points raised 

We look forward to receiving your revised manuscript.

Kind regards,

Claude Prigent

Academic Editor

PLOS ONE

Journal Requirements:

Reviewers' comments:

Reviewer's Responses to Questions

**Comments to the Author**

1. Is the manuscript technically sound, and do the data support the conclusions?

Reviewer #1: Partly

Reviewer #2: Yes

2. Has the statistical analysis been performed appropriately and rigorously? 

Reviewer #1: Yes

Reviewer #2: Yes

3. Have the authors made all data underlying the findings in their manuscript fully available?

Reviewer #1: Yes

Reviewer #2: Yes

4. Is the manuscript presented in an intelligible fashion and written in standard English?

Reviewer #1: Yes

Reviewer #2: Yes

5. Review Comments to the Author

Reviewer #1: Top3b is a unique topoisomerase known for its ability to interact with and resolve complexities in both DNA and RNA. Mutations in Top3b have been linked to various psychological disorders, but how these mutations affect RNA processing and contribute to disease phenotypes is not well understood. Teimuri and Suter investigated the roles of Top3b in RNA metabolism, and found some evidence suggesting that deletion of Top3b and its catalytic mutant (Y332F) are associated with aging, developmental and neurological defects, potentially due to misregulation of target transcripts using Drosophila model. The results should be of interest to the topoisomerase field, and also add new knowledge on how Top3b mutation may affect neurodevelopment, neurodeneration, and aging. However, while their study uncovers some new aspects of Top3b’s function in RNA metabolism, the authors need to perform some necessary experiments to improve the scientific quality of their data. The additional evidence is required to better support their conclusions and to meet standard of Plos One.

Major points

- The authors should provide concrete evidence to show that their newly made Top3b-mutant strains are correct. The authors should provide genomic DNA sequencing data for the mutant alleles. They also need to show Western Blotting for Top3b to confirm the null and normal levels of Top3b expression in Top3b26, Y332F-KI, and delta-RGG mutants, respectively. These experiments are critical controls to verify these Top3b-mutant strains.

- In Fig 1, the authors primarily utilized simple representations of their analysis lacking much information from their sequencing data. I suggest that the authors should incorporate more in-depth analysis techniques, such as volcano plots, dot distribution graphs, principal component analysis, etc. For instance, volcano plots for each mutant vs WT should be shown first so that readers can have an overall view (expression level and p-values) in the mutants. Then the venn graph can be followed.

- Add data points and mean values on all bar graphs.

- For the RIP assay, the authors immunoprecipitated GFP-tagged Top3b using GFP antibodies. To validate the covalent binding of RNA to the -OH of the Y332 residue, they utilized GFP-Top3b-Y332F as a negative control. The problem of this assay is that GFP-Top3b-Y336F protein still has the RGG-box capable of binding to mRNAs non-covalently. Therefore, the authors need to do necessary controls to rule out whether the observed binding is co-valent or non-covalent. Can authors do some RIP-qPCR under both denaturing and non-denaturing conditions for some candidates to verify their observed difference between WT vs. Top3b-Y336F mutant? Moreover, they should include the RIP-seq assay for GFP-Top3b-RGG deletion mutant, which could allow them to identify mRNAs bound through RGG-box. Furthermore, they need to include an additional negative control that does not express GFP (e.g. ORR or w1118 background flies for their fly lines). This control would help rule out any non-specific binding of the GFP antibodies and the beads.

- To analyze the relationship between Top3b binding and RNA levels, the authors should conduct a comparative analysis of RIP-seq and RNA-seq data. This involves examining the overlap between the targets identified in RIP-seq, which represent genes bound by Top3b, and the genes showing differential expression in RNA-seq. This comparison will allow to determine if there is a direct effect of Top3b binding on RNA levels.

- The authors discussed the possibility that Top3b may share the target transcripts with FMRP in discussion (Fig 7; they cited study done by Spradling group Science 2018). Top3b has been known to genetically and biochemically interact with Fmr1 (Xu 2013 NN). To investigate the functional and molecular roles of Top3b in collaboration with FMRP, it would be beneficial to compare its RNA targets with those of FMRP as the authors discussed. However, rather than referring to Science 2018, which is done in matured oocyte, I recommend utilizing data from Zhang et al. Nature Comms in 2022 (GEO dataset GSE143821). This dataset contains fine-scale timing RNA-seq data of Fmr1 mutant early embryos at different time points (0-0.5, 0.5-1, 1-2, 2-3 hours). By comparing the RNA targets of Top3b with those of Fmr1, they can assess any similarities or overlaps

- The authors stated that wild type Top3b and GFP tagged Top3b have similar expression level in Fig S5, but the supplementary materials do not have Fig S5 (has only Fig S1-4). The data may be in Table S5. A simple Western blotting with a proper loading control may be more accurate than mass spectrometry for this purpose.

Minor points

- On line 113-116, it is not clear how the authors determined the directly dependent on Top3b and obtained 315 RNAs. They need to clarify more details.

- In figure 2A, how reliable the association between two lists linked? Since the authors used embryos, I am not sure if the altered genes match with diseases linked genes. For example, does Fmr1 mutant embryo pick up many shared genes in AD or FXS? They need to validate if the method is reliable.

- In Figure 2A-B, the reliability of the association between the two lists of genes needs to be validated first. As the authors utilized embryos, it is unclear whether the altered genes identified in the study correspond to genes implicated in neuronal diseases. For instance, it would be essential to determine whether the genes identified in Fmr1 mutant embryos (see my major point above) overlap significantly with genes associated with AD or FXS.

- In Fig 3, show datapoints and average values. Change the label ‘wild-type’ embryonic RNA-IP to ‘GFP-Top3b’ embryonic RNA-IP or do not use ‘wild-type’. It is no longer wild-type.

- In Fig 3, the binding observed through the RIP appears relatively weak, as indicated by fold changes of less than 2(log2[2] = 4 fold). The authors should show qPCR validation of the RIP assay. For sequencing, I suggest utilizing DESEQ2 or other reliable programs to analyze the data and filter out non-significant p-values, thereby focusing on strong interactors.

- Show or address the common characteristics of the homologous targets (length, isoform numbers, cellular localization, etc) if there are any in Fig 3F

- In Fig 4, The authors need to include counterstain for non-Top3b targets for the specificity of the changed targets. Also, some samples, particularly Kst, may exhibit high background. The cellular localization of Kst during early embryogenesis is strictly limited to apical membrane whereas the wildtype image shows apical & lateral regions with high intensity background in the non-membrane regions (Thomas and Kiehart Development 1994). Also, the authors should include quantification of the signals.

- For all images, the authors provide high mag images. All images are taken with low magnification and difficult to assess signals.

- It is interesting that Y332F exhibits strong nuclear localization. This observation raises the possibility that Y332F may manifest a gain-of-function phenotype rather than a loss-of-function. Alternatively, the observed phenotypes could arise from mis-localization of interacting proteins such as Tdrd3 and Fmrp, particularly if they form stable complexes and localized at the nucleus.

- For Y332F, address and include Top3bY332F staining in other large tissues (nurse cell or salivary gland) to confirm nuclear localization is general phenomenon.

- I do not see where the authors obtained Shot, Dhc and bH-spectrin(kst) antibodies. Cite the providers properly in the methods.

- Figure S4 showed that Top3b-KO and Y336F mutant strains display shortened lifespan. The data are consistent with earlier findings in mouse by James Wang and colleagues. The data should be of general interest and included in a main figure.

- The Discussion may be too long and can be shortened. The authors may want to reduce the redundancy with the Results.

Reviewer #2: Major conclusions of this manuscript are that Top3β and its catalytic activity are required to produce a considerable number of RNAs during drosophila embryonic development and to avoid deleterious neuromuscular phenotype.

These conclusion extends prior reports showing that: 1/ Particularly long pre-mRNAs and RNAs with long ORFs and long UTRs depend on Top3β for their normal expression levels, 2/ Top3β suppresses R-loops can therefore promote transcription, 2/ Top3β can promote splicing and the production of mature mRNA/ splicing isoforms, and 3/ Top3β is associated with RNA transport and help their proper localization in large cells, 5/ Top3β can stabilize a subset of RNAs, and 6/ Top3β can also promote translation of bound/ target RNAs. The authors should clearly discuss these possible mechanisms and cite pertinent references in relevant sections of the Introduction, Results and Discussion.

The manuscript can be improved by addressing the following points:

1. Line 657-661: the same sentence is repeated twice.

2. The way the author present data in figure 1A is confusing. The authors should show Venn diagram displaying overlap between downregulated genes in Top3β26, Top3βY332F, and Top3βΔRGG with respect to wild type. The overlapping downregulated RNAs between Top3β26 and Top3βY332F should ideally represent RNAs directly dependent on Top3β for their embryonic expression/ localization in embryo. Rest of the downregulated RNAs in Top3β26 embryo should represent RNAs that do not directly depend on Top3β catalytic activity.

3. Figures 1B-D: please mention number of RNAs in each of the groups mentioned here (either in results section or in figure/ figure legend).

4. Could the TOP3B CHIP seq be done in nurse cells/ maternal cells. Particularly, what is overlap between the RNAs downregulated in Top3β26, Top3βY332F embryos and genes that are bound specifically to wild-type Top3β? If Top3β bound genes that are overlapping with the RNAs downregulated in Top3β26, Top3βY332F embryos are also long, then only one can conclude that Top3β and its catalytic activity have a strong effect on the transcription of long mRNAs.

5. Downregulated RNAs in Top3β26, Top3βY332F embryos can also have problems with RNA maturation, RNA transport and RNA stability. What is the overlap between RNAs downregulated in Top3β26, Top3βY332F embryos and RNAs that are bound specifically to wild-type Top3β? If these overlapping RNAs display features like long ORFs, long UTRs then one can conclude that Top3β and its catalytic activity have a strong effect on the production of normal expression levels of long pre-mRNAs and RNAs with long ORFs and long UTRs in the embryos.

6. The RNA immunoprecipitation is an important experiment for this study, yet the authors skipped over the issues of how pulled-down covalent TOP3B-RNA can be directly constructed into a sequencing library. A little more information on the generation of cDNA library will help the readers to better understand the scope and limitation of the method. Are random primers used?

7. The RNA immunoprecipitation was not carried out with the RGG-box deletion mutants, which would have provided some unique insights.

8. Line 281: The author concluded that the Shot protein signal levels and localization were less affected in the RGG-deleted mutant embryos, but the images in Figure 4 do not support this conclusion. Please clarify.

9. In the results section “RNAs depending on Top3β for their normal expression” please define the nature of mutation in the Top3β mutant Top3β26 after you introduce it first time in the second line. It will benefit the readers.

10. Results section “RNAs depending on Top3β for their normal expression”, last 5 lines: the way authors argued and defined 315 RNAs as directly dependent on TOP3β for their expression is not very clear. Authors should clarify this statement.

11. Supplementary Fig. 5 as mentioned in line 190 of result section “In vivo RNA interactions of Top3β in the absence of bulk transcription” not found within the submitted supplementary section. Please clarify.

12. Correlation between RNAs enriched in wild type Top3β IP compared to the Y332F mutant and their length (Fig. 3A) is not very strong (r=0.121). Please clarify “found that longer RNAs were indeed more likely to covalently bind to Top3β as they showed higher enrichment in the wild-type embryonic extracts (Fig. 3A)”.

13. Figure 4: why Shot protein expression is weaker in Top3βΔRGG mutant even though shot mRNA expression remains unaltered with respect to wild type embryo.

14. Authors should also find out the RNAs pulled down by Top3β-ΔRGG protein. This can clarify the function of RGG domain in Top3B catalytic activity.

6. PLOS authors have the option to publish the peer review history of their article (what does this mean? ). If published, this will include your full peer review and any attached files.

**Do you want your identity to be public for this peer review?** For information about this choice, including consent withdrawal, please see our Privacy Policy .

Reviewer #1: No

Reviewer #2: No

---

## [Author Response · Author response to Decision Letter 1]

13 Jul 2024

The responses have been uploded as a pdf file.

---

## [Decision Letter · Decision Letter 1]

2 Sep 2024

PONE-D-24-09977R1

RNA Targets and Physiological Role of Topoisomerase 3β

PLOS ONE

Dear Dr. Suter,

Thank you for submitting your manuscript to PLOS ONE. After careful consideration, we have decided that your manuscript does not meet our criteria for publication and must therefore be rejected.

Specifically:

The revised version of the manuscript has not been sufficiently improved to be considered for publication. 

Sorry I hope that the reviewers comments will help you improving your manuscript to be submitted to another journal.

I am sorry that we cannot be more positive on this occasion, but hope that you appreciate the reasons for this decision.

Kind regards,

Claude Prigent

Academic Editor

PLOS ONE

Reviewers' comments:

Reviewer's Responses to Questions

**Comments to the Author**

1. If the authors have adequately addressed your comments raised in a previous round of review and you feel that this manuscript is now acceptable for publication, you may indicate that here to bypass the “Comments to the Author” section, enter your conflict of interest statement in the “Confidential to Editor” section, and submit your "Accept" recommendation.

Reviewer #1: (No Response)

2. Is the manuscript technically sound, and do the data support the conclusions?

Reviewer #1: Partly

3. Has the statistical analysis been performed appropriately and rigorously? 

Reviewer #1: No

4. Have the authors made all data underlying the findings in their manuscript fully available?

Reviewer #1: No

5. Is the manuscript presented in an intelligible fashion and written in standard English?

Reviewer #1: Yes

6. Review Comments to the Author

Reviewer #1: In this version, the authors addressed some but not all of my requests in my previous review. The authors often claim that they do not agree with my specific comments and requests. As a result, the manuscript has limited improvement, and its major problems still remain. I therefore cannot recommend its publication.

Here are just some additional comments.

1. The authors did provide the sequencing data for their newly-generated Top3b mutant strains in their response letter. To me, these data are important to confirm that the new strains are correct. In my opinion, the data should be included in a Supplemental Figure.

2. To address my question whether the Top3b mutant proteins are expressed at similar levels to that of WT in newly-generated strains, the authors provided IP-Mass data for GFP-Top3b-WT and different mutant proteins. But these data are not what I am looking for. My question is whether endogenous Top3b-WT, Y336F and dRGG proteins are expressed at similar levels. For example, if the mutant proteins are expressed at lower levels than that of WT, one can argue that the observed phenotype in the mutant flies is due to reduced mutant protein, rather than due to loss of Top3b catalytic or RNA binding activity. The authors should examine the endogenous Top3b-WT and mutant protein levels in WT, KO, vs. Y336F, vs. dRGG flies by Western blotting. The authors claimed that they have RNA-seq data showing that Top3b RNA level is reduced in Top3b-KO, but not Y336F or dRGG mutants. But RNA levels do not equal to protein levels.

3. The authors have certainly improved the RNA-seq data presentation with the addition of new volcano plots and dot graphs, which provide a clearer and more comprehensive visualization of the data. However, I still have concerns regarding the data tables and their consistency with the figures. Specifically, there are discrepancies between the results presented in the figures and those in the supplementary tables. For example, in analyzing Table S1, there are 2,656 downregulated and 272 upregulated genes (log2>1, Adjp<0.05), whereas Fig 1a shows a much higher number of differentially expressed genes (DEGs). Similarly, Tables S2 and S3 show inconsistencies with the figures, which suggests that less stringent filtering may have been applied. This could be also due to annotation issues as the Supp tables contains protein coding genes, pseudogenes, transposons and other non-coding RNAs, and I believe the authors’ main focus is protein coding mRNAs. The authors should double-check their analysis and correct any discrepancies in the reported numbers.

4. The authors did not perform any experiments to rule out that the binding is due to covalent or non-covalent. Without such controls, I am not convinced their reported binding is due to the covalent cleavage complex, or non-covalent through RGG or other nucleic acid-binding motifs. I agree that GFP-Top3b-Y332F may be a control to reduce non-covalently bound RNAs. However, due to the relatively low fold enrichment values observed in the RIP-seq data (e.g., the strongest interaction show ~3-fold enrichment in Table S10, which is somewhat low for RNA-IP experiments), I still think that it is necessary to perform RIP-qPCR support their RIP-seq data and to validate conclusions of their study.

5. The authors claim that more than 4000 genes are altered more than 2-fold in Top3b-KO flies based on RNA-seq data. We are skeptical about such a large change, because the phenotype of Top3b-KO flies is not very strong. We have tried to download the RNA-seq data deposited at GEO by the authors to check the validity of author’s claims. However, these files appear to contain errors, which prevent us from analyzing these data. Possibly, the errors could be generated by disruption in the uploading process, which occur quietly commonly. Can the authors resubmit the their raw to GEO, and make sure that the uploaded files can be downloaded and analyzed?

7. PLOS authors have the option to publish the peer review history of their article (what does this mean? ). If published, this will include your full peer review and any attached files.

**Do you want your identity to be public for this peer review?** For information about this choice, including consent withdrawal, please see our Privacy Policy .

Reviewer #1: No

- - - - -

---

## [Author Response · Author response to Decision Letter 2]

7 Nov 2024

The file is also provided as a pdf. In the pdf, I used different colors for comments, responses and quotes. Please use the pdf. (Just in case, I pasted the file in here, too.)

6. Review Comments to the Author

Reviewer #1: In this version, the authors addressed some but not all of my requests in my previous review. The authors often claim that they do not agree with my specific comments and requests. As a result, the manuscript has limited improvement, and its major problems still remain. I therefore cannot recommend its publication.

I appreciate the constructive comments this reviewer gave us and we made changes to improve the manuscript. However, we refrained from performing experiments that would not add new value to the paper. I tried to explain our arguments already in the previous response file (added again in the detailed part below in green).

Here are just some additional comments.

1. The authors did provide the sequencing data for their newly-generated Top3b mutant strains in their response letter. To me, these data are important to confirm that the new strains are correct. In my opinion, the data should be included in a Supplemental Figure.

We have no problem doing this if the editor and journal request it (please consider that we already have numerous suppl. Figs. & Tables). I have now included this, but depending on the editorial decision, I could remove it again. Please let me know.

Our previous response was:

“==> We now provide more details in the Methods section, including at which steps the sequence of the mutant alleles was verified. We also deposited these strains at the stock center, where they are freely available (the order numbers are listed in our Methods section). Because the mutagenesis method has become standard and the strains are freely available, we do not see a need to show the primary sequencing data in the paper but appended it to our response for the reviewer’s perusal.”

2. To address my question whether the Top3b mutant proteins are expressed at similar levels to that of WT in newly-generated strains, the authors provided IP-Mass data for GFP-Top3b-WT and different mutant proteins. But these data are not what I am looking for. My question is whether endogenous Top3b-WT, Y336F and dRGG proteins are expressed at similar levels. For example, if the mutant proteins are expressed at lower levels than that of WT, one can argue that the observed phenotype in the mutant flies is due to reduced mutant protein, rather than due to loss of Top3b catalytic or RNA binding activity. The authors should examine the endogenous Top3b-WT and mutant protein levels in WT, KO, vs. Y336F, vs. dRGG flies by Western blotting. The authors claimed that they have RNA-seq data showing that Top3b RNA level is reduced in Top3b-KO, but not Y336F or dRGG mutants. But RNA levels do not equal to protein levels.

I fully understand the concern raised here. However, we have already shown in different ways that this is not the case. First of all, the reviewer’s concern applies mainly the Y332F mutant, the focus of this manuscript.

- From the stainings of the mutant embryos, where the same settings were used for the null mutant, the wild type, and the point mutants (Fig. 5, last column), it appears that the Y322F protein is expressed at levels indistinguishable from the ones of the wild-type protein.

- In this context, the fact that the Top3� RNA levels are down in the Top3�26 null mutant but not in the two point mutants (Supplementary Table S1-3) not only shows that the mutant RNA is expressed at the same level as the wild-type RNA, it is also consistent with the above interpretation of the Figure 5 results, which shows normal Top3� protein staining levels for the point mutants but no (or background) signal for Top3�26.

- We now know that the RNA is made at the same level, that it is translated and produces a stable protein. With this knowledge, we maintain that we can address the quantitative aspects of the protein expression and stability using the quantitative MS (qMS) assay with the GFP-tagged protein (Supplementary Table S9). In the criticism, Reviewer 1 still insists on a Western blot instead of “MS” (in the second review, the reviewer mentioned only “MS”, overlooking or ignoring the fact that it was quantitative MS). Quantitive MS analysis, done properly, is at least as good as Western blotting.

(For the delRGG mutant, which is not the focus of this manuscript, we have the following arguments:

- Genetically, it behaves way more like the wild type at the stages we examined. It cannot be a null

or close to a null mutant.

- RNA levels are also normal

- qMS data shows that the tagged protein is expressed normally

- protein staining of embryos shows reduced staining in the nuclei, but not in the cytoplasm where

the reported interactions take place (there is virtually no transcription at this stage)).

Here is what we responded previously:

=> The quantitative mass spectrometry (qMS) results presented in the Supplementary Table S5 are normalized to the eGFP-only expression done in parallel (and GFP is the target of the antibody). The log2 fold-change (compared to eGFP alone) is almost the same between Top3β+::eGFP (4.7) and the mutant versions (4.6 for delRGG and 4.4 for Y332F) (lane 2 in the Supplementary Table S9). This result serves the same purpose as a Western blot. We now mention this result also in the Methods section. It is also consistent with the results of the embryonic stainings (Fig. 5) which were done with background controls and the same laser power and confocal microscope settings.

That Top3b26 mRNA levels are down in the null mutant, we mention in the paper and refer to the quantification by RNAseq obtained by the analysis of its transcriptome (Supplementary Table S1; and S2-3 for the other mutants): Here are the values: for Top3B26: 26/wt: 6.75 E50 (adjpv) / -2.32 (log2fc); for Top3BY332F: wt/Y332F: 0.33 (adjpv) / -0.19 (log2fc); for Top3B deltaRGG: wt/delRGG: 0.17 (adjpv) / -0.28 (log2fc). The “point” mutants do not show a reduction (but maybe a slightly higher mRNA expression level).

3. The authors have certainly improved the RNA-seq data presentation with the addition of new volcano plots and dot graphs, which provide a clearer and more comprehensive visualization of the data. However, I still have concerns regarding the data tables and their consistency with the figures. Specifically, there are discrepancies between the results presented in the figures and those in the supplementary tables. For example, in analyzing Table S1, there are 2,656 downregulated and 272 upregulated genes (log2>1, Adjp<0.05), whereas Fig 1a shows a much higher number of differentially expressed genes (DEGs). Similarly, Tables S2 and S3 show inconsistencies with the figures, which suggests that less stringent filtering may have been applied. This could be also due to annotation issues as the Supp tables contains protein coding genes, pseudogenes, transposons and other non-coding RNAs, and I believe the authors’ main focus is protein coding mRNAs. The authors should double-check their analysis and correct any discrepancies in the reported numbers.

I appreciate that this issue was pointed out. The discrepancy was due to using rounded (log2fc) values and log2fc ≥1 in one case and the more precise log2fc >1 in the other. In the new version, we now use log2fc >1 for both (without arbitrary rounding). Additionally, because I see less value in using DEGs, I reanalyzed the data and presented down- and upregulated genes separately in Fig 1A using <-1 and >1 values instead of ≤ and ≥.

This is the new version of Fig. 1A:

4. The authors did not perform any experiments to rule out that the binding is due to covalent or non-covalent. Without such controls, I am not convinced their reported binding is due to the covalent cleavage complex, or non-covalent through RGG or other nucleic acid-binding motifs. I agree that GFP-Top3b-Y332F may be a control to reduce non-covalently bound RNAs. However, due to the relatively low fold enrichment values observed in the RIP-seq data (e.g., the strongest interaction show ~3-fold enrichment in Table S10, which is somewhat low for RNA-IP experiments), I still think that it is necessary to perform RIP-qPCR support their RIP-seq data and to validate conclusions of their study.

I tried to explain why this request does not make sense to me. The only difference between the protein that can bind the RNA covalently and the control is the one -OH group that makes to covalent bond. Of course, RNAs are known to bind to the RGG motives (and possibly other regions, too), but these are non-covalent interactions that will be washed off by SDS. I have added the following sentence to the description of the method to bring this point across: “…because RNAs non-covalently interacting with Top3β will be washed off the enzyme by the ionic detergent.” Of course, there will still be background levels of mRNAs that non-covalently stick to the protein, but these will be present in the control, too, and at the same “background level” in the IP with the functional enzyme and the enzymatically dead one. Therefore, they will be identified as non-specific backgrounds.

We expected that the true enrichment cannot be very high because the covalent phase is only part of the enzymatic cycle and only a fraction of an RNA species interacts with the enzyme at a given time (we pointed this out already, but elaborated a bit more on this on p12/13). Please note that this is much different from an RNA-binding protein that sits continuously on an mRNA because only a small fraction of a specific RNA pool is bound to Top3B at one time. (On the other hand, theoretically, every expressed mRNA could become a Top3B target.)

Furthermore, RIP-qPCR relies on the same sample preparation as the RIP-seq experiments that we had performed in biological triplicates. Also, variations of the RIP protocol (e.g., artificial cross-linking) would prevent comparison with our presented dataset. Trying to use our resources efficiently, it seemed to make much more sense to further test the value of the results by testing for the predicted effects on the encoded proteins and the cellular significance. (Without effects on them, further studies would be difficult to justify). This is why we used this different, independent test setup. The results showed clearly that Top3B affects mRNAs and proteins encoded by the identified target mRNAs (Fig. 5). This approach should be more valuable for researchers interested in studying individual Top3B targets than performing the same IPs in more than three biological replicates.

=> We do not see the value of such an experiment. Binding to the RGG motif does not involve covalent interactions. These weaker interactions are generally not stable under our conditions with the ionic detergent SDS. Furthermore, unexpectedly strong interactions with the RGG motif would show the same binding to RGG in the mutant control and the wild type. It would, therefore, show up as a false negative and this is less of a problem than a false positive (because we found enough targets). Please also note that the very high number of different samples (different genotypes, different conditions, and all x3) forced us to focus on the most important controls, which we still think we did.

5. The authors claim that more than 4000 genes are altered more than 2-fold in Top3b-KO flies based on RNA-seq data. We are skeptical about such a large change, because the phenotype of Top3b-KO flies is not very strong. We have tried to download the RNA-seq data deposited at GEO by the authors to check the validity of author’s claims. However, these files appear to contain errors, which prevent us from analyzing these data. Possibly, the errors could be generated by disruption in the uploading process, which occur quietly commonly. Can the authors resubmit the their raw to GEO, and make sure that the uploaded files can be downloaded and analyzed?

(Without rounding the data, I now found 2680 genes downregulated in the null mutant (using the less stringent criteria and 1453 with the more stringent ones) according to Fig. 1A). It should be kept in mind that we expect that every RNA can become a Top3B target and, therefore, every gene could already be directly affected. Also, as opposed to the on-off effects one often sees in studying developmental control processes, the reduction of RNA levels can be expected to be less drastic because we can expect that only a fraction of a specific mRNA pool depends on Top3B (e.g., the ones that run into structural problems). Furthermore, it was estimated that in Drosophila, more than half the genes can be knocked down without displaying easily detectable phenotypes and it has been shown that heterozygous deficiencies are mostly viable, suggesting that half the dose of gene products from the deficiency region causes no serious problems in a lab setting. The combination of these three points probably explains why the phenotypes are not more pronounced. I am not “skeptical” about these numbers, I think they fit the process we describe in this manuscript perfectly.

Our Bioinformatics support group usually submits such data to a different database. It was only because this reviewer requested that we submit it to this database. Strangely, the quality control of GEO did not detect the problem. We have now submitted the data again.

There is no reviewer 2 this time.

Reviewer 2 was more positive and had more minor requests and questions in the first round. Did we satisfy this reviewer with our revisions?

---

## [Decision Letter · Decision Letter 2]

11 Dec 2024

PONE-D-24-09977R2RNA Targets and Physiological Role of Topoisomerase 3βPLOS ONE

Dear Dr. Suter,

Thank you for submitting your manuscript to PLOS ONE. After careful consideration, we feel that it has merit but does not fully meet PLOS ONE’s publication criteria as it currently stands. Therefore, we invite you to submit a revised version of the manuscript that addresses the points raised during the review process.This is your last chance, please revise your manuscript in the light of the reviewers' comments. In particular, please pay attention to each comment/question from reviewer #4, as they raise important issues.

Reviewer 4 gives you advice on the structure of the manuscript, I agree with his remarks. These are points which might seem minor but which ultimately improve the logic of the reasoning and make for more enjoyable reading.

We look forward to receiving your revised manuscript.

Kind regards,

Claude Prigent

Academic Editor

PLOS ONE

Journal Requirements:

1. We notice that your supplementary figures are uploaded with the file type 'Figure'. Please amend the file type to 'Supporting Information'. Please ensure that each Supporting Information file has a legend listed in the manuscript after the references list.

Additional Editor Comments (if provided):

Reviewers' comments:

Reviewer's Responses to Questions

**Comments to the Author**

1. If the authors have adequately addressed your comments raised in a previous round of review and you feel that this manuscript is now acceptable for publication, you may indicate that here to bypass the “Comments to the Author” section, enter your conflict of interest statement in the “Confidential to Editor” section, and submit your "Accept" recommendation.

Reviewer #3: (No Response)

Reviewer #4: (No Response)

2. Is the manuscript technically sound, and do the data support the conclusions?

Reviewer #3: Yes

Reviewer #4: Partly

3. Has the statistical analysis been performed appropriately and rigorously? 

Reviewer #3: Yes

Reviewer #4: Yes

4. Have the authors made all data underlying the findings in their manuscript fully available?

Reviewer #3: Yes

Reviewer #4: Yes

5. Is the manuscript presented in an intelligible fashion and written in standard English?

Reviewer #3: Yes

Reviewer #4: Yes

6. Review Comments to the Author

Reviewer #3: The RNA topoisomerase activity of Top3B in eukaryotes is an interesting topic, and have implications for neurological development and other physiological functions that potentially require such RNA topoisomerase activity. The identification of RNA targets of Drosophila Top3B in early embryonic stage is informative. The alteration of subcellular localization of certain gene products in an active site mutant of Top3B is an important result, suggesting that large genes and long and complex transcripts may need Top3B to be transported and translationally controlled.

Both Ref 74 and data in Fig 1A of this manuscript showed that the active site Tyr point mutation that specifically disrupted Top3B catalytic activity only partially recapitulated the effects of TOP3B-null mutation. Su et al. suggested that Top3B–TDRD3 can regulate mRNA translation and turnover by mechanisms that are dependent and independent of topoisomerase activity. It would be useful for the authors to comment on the possibility of catalytic activity independent mechanism of RNA regulation by Top3B based on the Drosophila data comparing effect of Top3B26 and Top3BY322F mutations described here.

Reviewer #4: Major comments:

1. The authors show that TOP3B binds long maternal RNAs in a catalytic activity dependent manner in 0-2h embryos. While the absence and mutation of TOP3B leads to severe downregulations of mRNA abundance found, particularly those related to morphogenesis, synapse signaling, transmembrane transport, cell adhesion and cuticle formation. Yet this downregulation in maternal RNA abundance found in embryos does not manifest in developmental phenotypes.

As this transcriptomic analysis captures maternal mRNA from the oocyte and nurse cells, all these findings concretely suggest is that maternal mRNA, or at least large quantities of maternal mRNA is not necessary for embryonic and larval development. Overall, the argument for the role of TOP3B in neurodegeneration is weak as transcriptome analysis was only done in embryos.

2. While the authors further confirm TOP3B favors binding to long RNAs in Figure 3, this is neither a novel phenomenon nor is it important in neural development and development overall. It is also confusing when the authors mention that 0-2h embryos reflect the summation of changes in mRNA level from transcription in the oocyte/nurse cells to translation in the embryos (Line 171-173), while also using the lack of zygotic transcription of 0-2h embryos to isolate direct RNA targets of TOP3B (Line 245-246). The authors also reason that TOP3B is especially necessary for long RNAs that are prone to physical stress to their structure (Line 280-282). With this RNA-IP, do the authors mean to imply that TOP3B dominantly affects gene expression through binding to long RNA to facilitate its localization and translation with negligible effects on DNA and transcription?

3. Section on parallels between Drosophila TOP3B/FMR1 and mammalian TOP3B/FMRP does not logically flow with the previous section, and is almost negligible to the manuscript, placing it at the end would be more suitable to understate its low impact to the overall message.

4. Line 364-371 is an excuse with barely any relevance to both the previous section and the next paragraph, which only serves to remind the readers how insignificant TOP3B’s role is in embryogenesis and highlights the futility of visualizing RNA localization in embryos and not in neuronal cells. Placing Figure 5 data right after Figure 3 would improve readability.

5. Figure 6 demonstrates that TOP3B depletion and catalytic mutant finally have phenotypes in age-related degeneration. Yet all explanation behind the mechanism is based on an oocyte transcriptome, and other correlative clues (Figures 2 and 4) to neurodegeneration. The authors should consider a transcriptome analysis of young adult and aged animals to directly show the causative effects of the lack/mutation of TOP3B.

6. Finally, the authors introduced G4C2 repeats found in human neurodegenerative diseases to TOP3B fly mutants. The authors need to justify a few things: whether or not TOP3B has been shown to suppress toxicity of G4C2 repeats in Drosophila/other species and why the adult eyes specifically was used to assess a motor neurodegenerative disorder stimulated by G4C2 repeats.

Minor comments:

1. All titles (main title, result and discussion section titles) are vague and inconclusive.

2. Figure 2 and its text section is misplaced and irrelevant to the previous section, as there is nothing in the previous data (Figure 1, Supplementary Figures 1-3) reflecting a relationship between the transcriptome results with neurological disorders and cancer. Figure 2 only becomes relevant when grouped with Figure 6.

7. PLOS authors have the option to publish the peer review history of their article (what does this mean? ). If published, this will include your full peer review and any attached files.

**Do you want your identity to be public for this peer review?** For information about this choice, including consent withdrawal, please see our Privacy Policy .

Reviewer #3: No

Reviewer #4: No

---

## [Editor Report · Decision Letter 3]

13 Jan 2025

Drosophila Topoisomerase 3β binds to mRNAs in vivo, contributes to their localization and stability, and counteracts premature aging

PONE-D-24-09977R3

Dear Dr. Suter,

We’re pleased to inform you that your manuscript has been judged scientifically suitable for publication and will be formally accepted for publication once it meets all outstanding technical requirements.

Kind regards,

Claude Prigent

Academic Editor

PLOS ONE
---

## [Editor Report · Acceptance letter]

PONE-D-24-09977R3

PLOS ONE

Dear Dr. Suter,

I'm pleased to inform you that your manuscript has been deemed suitable for publication in PLOS ONE. Congratulations! Your manuscript is now being handed over to our production team.

Kind regards,

on behalf of

Dr. Claude Prigent

Academic Editor

PLOS ONE